# Ultrafast fMRI reveals serial queuing of information processing during multitasking in the human brain

Qiuhai Yue [1,2] ✉, Allen T. Newton[3] & René Marois [2,4,5] ✉

The human brain is heralded for its massive parallel processing capacity, yet influential cognitive models suggest that there is a central bottleneck of information processing distinct from perceptual and motor stages that limits our ability to carry out two cognitively demanding tasks at once, resulting in the serial queuing of task information processing. Here we used ultrafast (199 ms TR), high-field (7T) fMRI with multivariate analyses to distinguish brain activity between two arbitrary sensorimotor response selection tasks when the tasks were temporally overlapping. We observed serial processing of task-specific activity in the fronto-parietal multiple-demand (MD) network, while processing in earlier sensory stages unfolded largely in parallel. Moreover, the MD network combined with modality-specific motor areas to define the functional characteristic of the central bottleneck at the stage of response selection. These results provide direct neural evidence for serial queuing of information processing and pinpoint the neural substrates undergirding the central bottleneck.

The human brain, with its 87 billion neurons and thousands of connections per neuron, is heralded for its ability to process vast amount of information in parallel[1]. However, while that capacity is apparent at the micro- and meso-scale levels of functional brain organization, that may not be so at the whole-brain systems level supporting cognition. A severe limitation in processing information in parallel surfaces when we try to carry out two cognitively demanding tasks at once. Under such conditions, at least one if not both of the tasks suffer in performance[2–6]. The classic paradigm to investigate multi-tasking limitations is the psychological refractory period (PRP), in which two sensory stimuli, each paired with an arbitrary motor response, are presented at various intervals from one another. At short intervals, the second of the two responses is significantly delayed while the response time to the first task is largely unaffected, whereas little or no Task 2 delay is observed at longer intervals (Supplementary Fig. 1). This PRP effect is robust, occurring even when tasks do not overlap in sensory or motor modalities and with a wide range of cognitive processes, including response selection, working memory, and mental

rotation[6–16]. Considerable evidence suggest that this "bottleneck" occurs at the central, amodal information processing stage of response selection that can only act on one task at a time[5,6,17,18], leading to the postponement of Task 2 processing until the first task is processed, in contrast to perceptual and response execution stages that can process modality-specific information largely in parallel (Supplementary Fig. 1). Such serial queuing at a central stage of information processing is not strategic, for it takes place even under conditions in which parallel processing is more efficient and financially incentivized[19,20]. Rather, it is thought to reveal an intrinsic inability for a general-purpose learning processor to simultaneously multiplex[21,22].

While there is considerable behavioral evidence for a serial mode of information processing in human cognition, the neural instantiation of this bottleneck—where and when the information flow switches from a parallel mode of information processing to serial queuing as postulated by classic cognitive models—has not yet been elucidated mostly owing to technical limitations. While several groups have used innovative approaches to tackle that question[7,8,12,23–27], none have had

[1]School of Psychology, Shenzhen University, Shenzhen, China. [2]Department of Psychology, Vanderbilt University, Nashville, TN, USA. [3]Vanderbilt University Medical Center, Nashville, TN, USA. [4]Vanderbilt Vision Research Center, Vanderbilt University, Nashville, TN, USA. [5]Vanderbilt Brain Institute, Vanderbilt University, Nashville, TN, USA. ✉e-mail: qyue@szu.edu.cn; rene.marois@vanderbilt.edu

the requisite combined spatial and temporal resolution necessary to track task-specific activity in dual-task conditions as each courses through the human brain. Using a combination of approaches—a task design that lengthens the cognitive stages of information processing and eschews overlaps in sensorimotor modalities, an ultrafast high-field method of functional image acquisition that is an order of magnitude faster than standard protocols, and multivariate analyses that tease apart task-specific brain activity—here we were able to track the flow of activity from each task as they course through the human brain from sensation to response execution in conditions in which capacity limits of information processing are not due to sensory or motor limitations. The results revealed clear evidence of serial queuing of amodal information processing in a specific set of parieto-frontal areas corresponding to the multiple-demand (MD) network, a network well known to undergird effortful performance of arbitrary cognitive tasks[28-31]. Moreover, we show that this MD network works in consortium with modality-specific motor areas to form a central bottleneck of information processing while leaving perceptual stages of information processing largely unaffected.

## Results

### Behavioral results

The behavioral paradigm consisted of two arbitrary sensory-response mapping tasks—one involved choosing the appropriate oculomotor response to a specific auditory stimulus (AO task) and the other required selecting the suitable manual response to a distinct visual color stimulus (VM task)—with each of the two tasks consisting of eight alternative sensory-response mappings (Fig. 1). These single-task trials were pseudo-randomly intermixed with dual-task trials while 26 subjects were imaged in a high-field (7T) scanner. As expected for response selection tasks with such number of alternatives[7], the response times under single-task conditions hovered around 1 s (Fig. 1d, e). To examine the response time costs under dual-task conditions, we compared these response times when the two sensory stimuli were presented at a short stimulus-onset-asynchrony (SOA) of 300 ms—short enough to engender dual-task costs in the second of the two tasks—and at a long SOA of 1500 ms, long enough to avoid dual-task costs (Supplementary Fig. 1c, d). Combining across task orders as the results were similar (Fig. 1d, e), we observed that the response times of the second of the two tasks in dual-tasks were significantly longer at the short SOA relative to the long SOA ($p = 1.13 \times 10^{-18}$) or to the single-tasks ($p = 6.03 \times 10^{-16}$), amounting to a PRP of over half a second (see Supplementary Fig. 2 for accuracy performance). By contrast, the dual-task costs to the first task (T1) were minimal (32 ms, short SOA T1 vs. long SOA T1; $p = 0.02$; short SOA T1 vs. single, $p = 0.1$). As expected, these data provide clear behavioral evidence for serial processing of Task 1 and Task 2 under dual-task conditions.

### Isolation of sensory, amodal central, and motor response areas

The initial analytical step towards assessing task-specific activity across stages of information processing consisted in isolating brain regions corresponding to putative sensory, central amodal, and motor response stages of information processing using univariate fMRI analyses. In the first GLM analysis, sensory and motor regions were identified by directly contrasting activity across the two single sensorimotor tasks, each first masked by their activation over baseline, using stimulus onset regressors for sensory ROI isolations and response timing regressors for motor ROI isolations (see "Methods"). This strategy led to auditory activation centered in the Herschel gyrus, corresponding to the primary auditory cortex (Fig. 2a and Supplementary Table 1), and visual activation in the posterior occipital cortex, corresponding to putative V1-V4 areas (Fig. 2b and Supplementary Table 1). A similar strategy was employed to isolate motor response areas. However, as the resulting activation for the manual response encompassed both somatomotor and

somatosensory areas, an anatomical mask was applied to the pre-central gyrus to limit the activation foci to purely motor regions in an area corresponding to the primary motor cortex for manual responses (Fig. 2d, Supplementary Table 1 and see also Supplementary Fig. 3). The oculomotor activation centered around the frontal eye field (FEF) in the dorsal frontal cortex (Fig. 2c, Supplementary Table 1 and see also Supplementary Fig. 3) and in the superior occipital/posterior parietal cortex. The latter contrast also led to a broad swath of activation in the visual cortex, owing to eye movements across the visual scene during the AO task[32-35].

To isolate central amodal brain regions, we reasoned in the second GLM analysis that such areas should be conjointly activated across both tasks[7,8,12,36]. Instead of using stimulus or response onset regressors, here we used whole event regressors whose durations were independently estimated from an a priori defined anatomical area (see "Methods"). The conjunction maps of the two task vs. baseline revealed activation foci in five cortical areas (Fig. 2e and Supplementary Table 2); the inferior frontal gyrus (IFG), the anterior insula (AI), the posterior dorso-lateral prefrontal cortex (pdPFC) slightly anterior and medial to the FEF (see also Supplementary Fig. 3), the medial frontal gyrus/dorsal anterior cingulate cortex (mFG/dACC), and the intra-parietal sulcus (IPS) that correspond closely to the core multiple-demand (MD) network (also known as the task positive network; see ref. 37), an ensemble of brain regions thought to undergird general-purpose cognitive operations in the service of ongoing task demands[28-30]. Additional activation foci in the conjunction SPMs were found in subcortical areas and the TPJ (see Supplementary Fig. 4 and Supplementary Table 2). We also directly contrasted dual-task trials to single-task trials to determine whether there were any dual-task specific brain regions specifically activated under dual-task conditions[38,39], taking care in the GLM to assign longer estimated responses to the dual-tasks relative to the single-tasks due to the compound nature of their events (see "Methods"). Replicating our previous findings, we found no such areas in frontal or parietal lobes[7,8,12] (see Supplementary Fig. 5).

To ascertain the sensory, motor, and central amodal characteristics of the regions of interest (ROIs) isolated in the single-task SPMs, we evaluated in a third GLM analysis their BOLD response timecourses in both single-task and dual-task conditions (see "Methods"), as these timecourses should exhibit distinct activation profiles in amplitude and in latency across the processing stages.

As expected of a purely sensory area, the auditory cortex ROI showed no activation to the single visual-manual task but a robust hemodynamic response in the auditory-oculomotor task (Fig. 3a). Importantly, this response was delayed by about 1.5 s (Peak latency = 5.045 s for the long-SOA VMAO task, or L-VMAO vs. 3.545 s for single-AO, $t(25) = 53.46$, $p = 2.7 \times 10^{-27}$, Cohen's $d = 10.48$; L-VMAO−1.5 s vs. single-AO, $p = 0.99$, BF(01) = 4.8; Onset latency = 2.656 s for L-VMAO vs. 1.267 s for single-AO, $t(25) = 12.99$, $p = 1.3 \times 10^{-12}$, Cohen's $d = 2.55$; L-VMAO−1.5 s vs. single-AO, $p = 0.3$; BF(01) = 3.0) when the auditory stimulus was presented second at long (1500 ms) SOA. Remarkably, we could still observe a similar, albeit smaller, shift in the response at the short (300 ms) SOA when the auditory stimulus was presented second (Peak Latency = 3.895 s for the short-SOA VMAO task, or S-VMAO vs. 3.545 s for single-AO, $t(25) = 12.48$, $p = 3.1 \times 10^{-12}$, Cohen's $d = 2.45$; S-VMAO−0.3 s vs. single-AO, $p = 0.09$, BF(01) = 1.2; Onset Latency = 1.541 s for S-VMAO vs. 1.267 s for single-AO, $t(25) = 3.39$, $p = 0.002$, Cohen's $d = 0.66$; S-VMAO−0.3 s vs. single-AO, $p = 0.7$; BF(01) = 4.6), not only in the curve-fit data but also in the raw averaged data (Fig. 3a). Importantly, no such activity delay was observed when the auditory task was presented first, either at short or long SOAs (Peak latency = 3.580 s for S-AOVM vs. 3.545 s for single-AO, $p = 0.2$, BF(01) = 2.2; 3.544 s for L-AOVM vs. 3.545 s for single-AO, $p = 0.96$, BF(01) = 4.8; Onset latency = 1.276 s for S-AOVM vs. 1.267 s for single-AO, $p = 0.9$, BF(01) = 4.8; 1.297 s for L-AOVM vs. 1.267 s for single-AO,

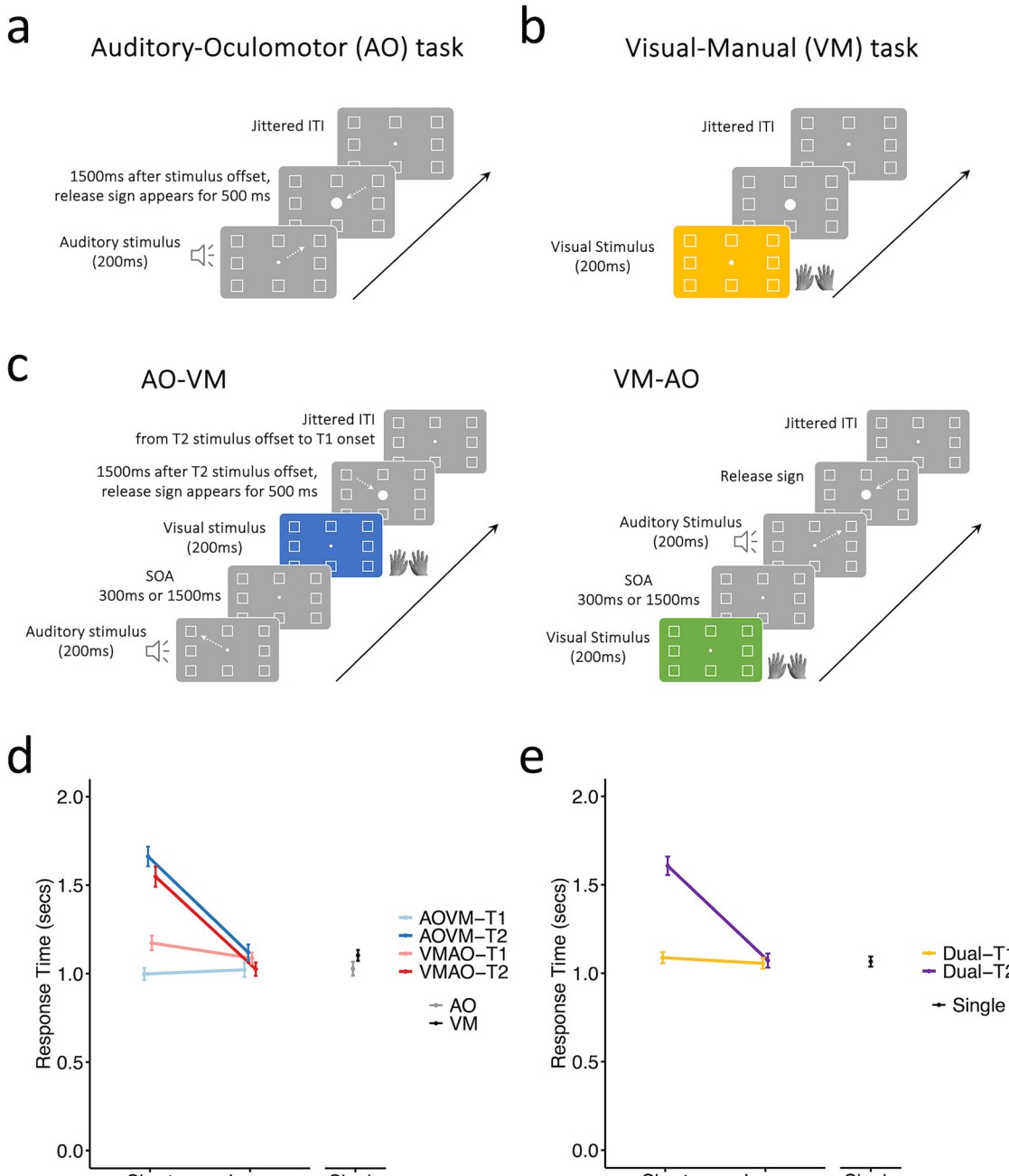

**Fig. 1 | Experimental design and behavioral results. a** In the Auditory-Oculomotor (AO) task, participants responded to one of eight auditory sounds by moving their eyes from a central fixation to one of eight possible target positions (marked by place holders) and maintaining the target eye position until a release stimulus appeared 1500 ms later, instructing them to return eye gaze to the central fixation point; (**b**) In the Visual-Manual (VM) task, participants responded to one of eight visual colors by pressing one of eight possible buttons, each assigned to a specific finger except for the two thumbs; (**c**) In the dual-task conditions, the order of the two tasks (AO-VM or VM-AO) and stimulus-onset asynchrony (SOA, 300 msec or 1500 msec) were varied across trials; (**d**) In-scanner behavioral results show response times for the single-task and dual-task conditions: In AOVM trials, RT2 = 1662 ms at short SOA vs. 1120 ms at long SOA, $t(25) = 21.49$, $p = 1.23 \times 10^{-17}$, $N = 26$,

paired-sample $t$-test, two-tailed; in VMAO, RT2 = 1548 ms at short SOA vs. 1025 ms at long SOA, $t(25) = 15.17$, $p = 4.04 \times 10^{-14}$, Long SOA RTs were not different from single-task RTs (long SOA AOVM RT2 vs. VM RT = 1103 ms, $t(25) = 0.57$, $p = 0.57$; long SOA VMAO RT2 vs. AO RT = 1028 ms, $t(25) = 0.22$, $p = 0.83$); (**e**) Response times for the single-task and dual-task conditions combined across two task orders. In dual-task conditions, RT2 = 1607 ms at short SOA vs. 1072 ms at long SOA, $t(25) = 23.75$, $p = 1.13 \times 10^{-18}$, $N = 26$, paired-sample $t$-test, two-tailed. The Long SOA RT2 was not different from Single-Task RT = 1066 ms (long SOA RT2 vs. single, $t(25) = 0.36$, $p = 0.72$). Long SOA RT1(1056 ms) was not different from single RT, $t(25) = 1.56$, $p = 0.13$). Error bars represent the standard error of the mean. Source data are provided as a Source Data file.

$p = 0.5$, BF(01) = 3.9, Fig. 3a). Also, as expected of a modality-specific sensory area, there were no difference in amplitude of activation among the five task conditions involving auditory sensory processing ($F(4,100) = 0.47$, $p = 0.76$, partial $\eta^2 = 0.02$, one-way repeated measures ANOVA, Fig. 3a). The timecourses in the visual cortex ROIs

(sampled across V1–V4) showed not only a response in the visuo-manual task, but also in the auditory-oculomotor task (Fig. 3b), consistent with the broad visual cortex activation observed in the SPMs (Fig. 2c). That AO-related activity in visual cortex was delayed by well over a second relative to the VM task (Onset latency = 2.119 s

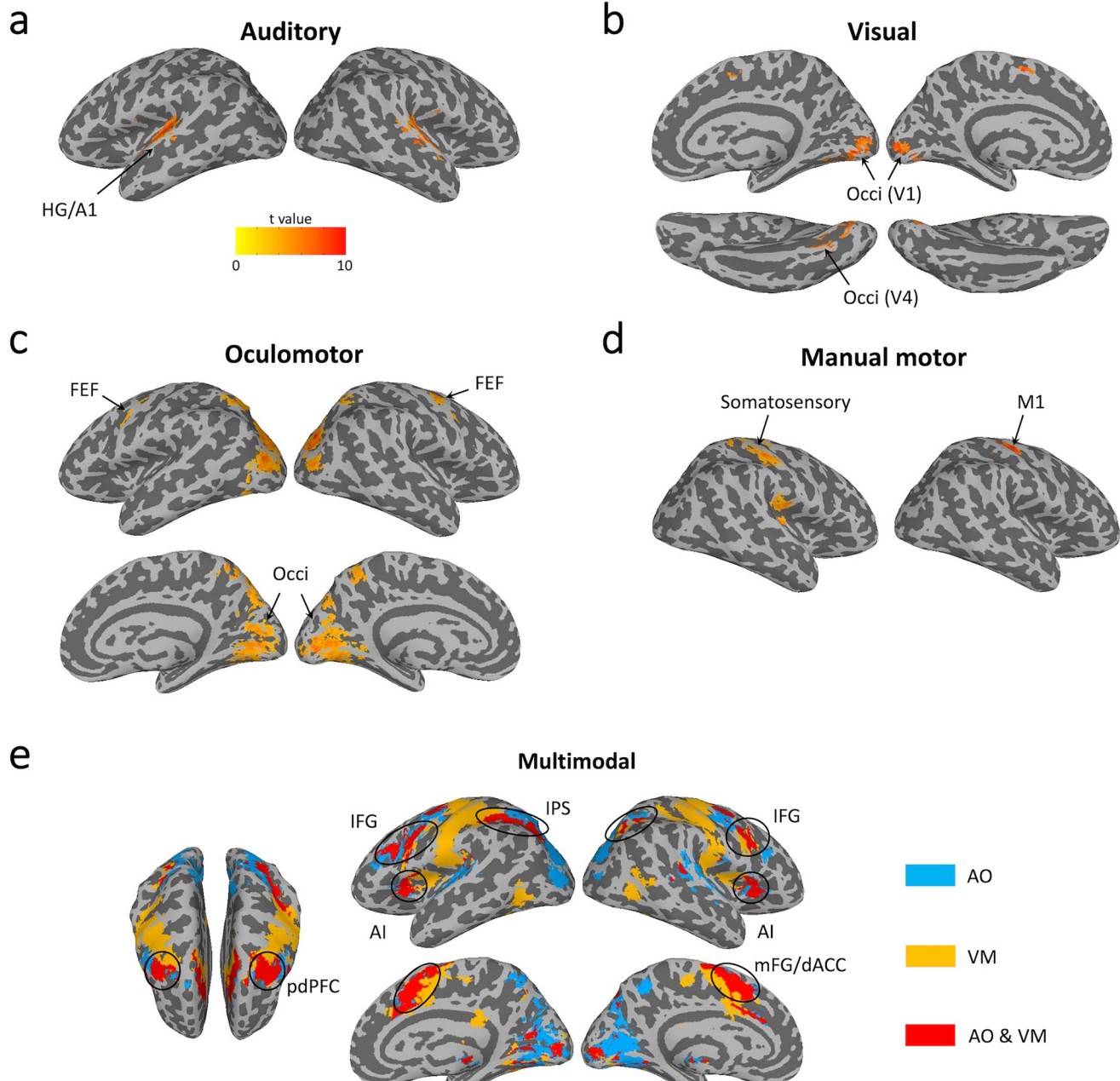

**Fig. 2 | Sensory, motor and multimodal regions of interest. a** Sound presentations in the single AO task activated auditory cortex around the bilateral HG/A1; (**b**) Color presentations in the single VM task activated visual cortex centered in bilateral putative V1–V4; (**c**) Moving eyes to target positions in the single AO task activated bilateral FEF as well as a large cluster covering bilateral parietal and occipital areas (including V1); (**d**) Pressing buttons in the visual VM task activated bilateral precentral (including M1) and postcentral gyri, as well as bilateral Rolandic areas. The sensory and motor ROIs are reported based on a threshold at voxel level of $p < 0.001$, and corrected at cluster level $\alpha < 0.01$, with cluster size >55 voxels.

**e** The conjunction analysis revealed activations in bilateral IFG, bilateral pdPFC, bilateral IPS, bilateral AI and bilateral mFG/dAC, corresponding to the multiple demand (MD) network. The conjunction ROIs are reported based on a threshold voxel level of $p < 0.001$, corrected at cluster level $\alpha < 0.01$, with cluster size >62 voxels. HG Heschl's gyri, A1 primary auditory cortex, FEF frontal eye field, M1 primary motor cortex, IFG inferior frontal gyri, pdPFC posterior dorsal prefrontal cortex, IPS intraparietal sulcus, AI anterior insula, mFG/dACC medial frontal gyri/dorsal anterior cingulate cortex, AO auditory-oculomotor, VM visual-manual.

for single-VM vs. 3.872 s for single-AO, $t(25) = 7.55$, $p = 6.6 \times 10^{-8}$, Cohen's $d = 1.48$; Peak latency = 4.306 s for single-VM vs. 6.258 s for single-AO, $t(25) = 7.27$, $p = 1.3 \times 10^{-7}$, Cohen's $d = 1.43$), as would be expected of a brain activity related to an eye movement to the target position that occurs a second later (relative to stimulus presentations) and that is followed 500 ms later by the appearance of the release sign and subsequent re-orienting to the central fixation, all events that would stimulate visual cortex activation. As we shall see below, this large oculomotor-related activation precludes the visual

cortex ROI from serving as a VM task-specific sensory area in the multivariate analyses.

Central, amodal areas involved in both sensorimotor tasks should demonstrate a distinct activation profile to the sensory areas, namely similar onsets of activation across both single-and dual-task conditions. In addition, they should show both stronger and extended activations in dual-tasks relative to single-tasks, owing to the prolongation of neural activity in these regions under dual-task conditions (see modeled activation in Supplementary Fig. 6; refs. 7,8,12). The

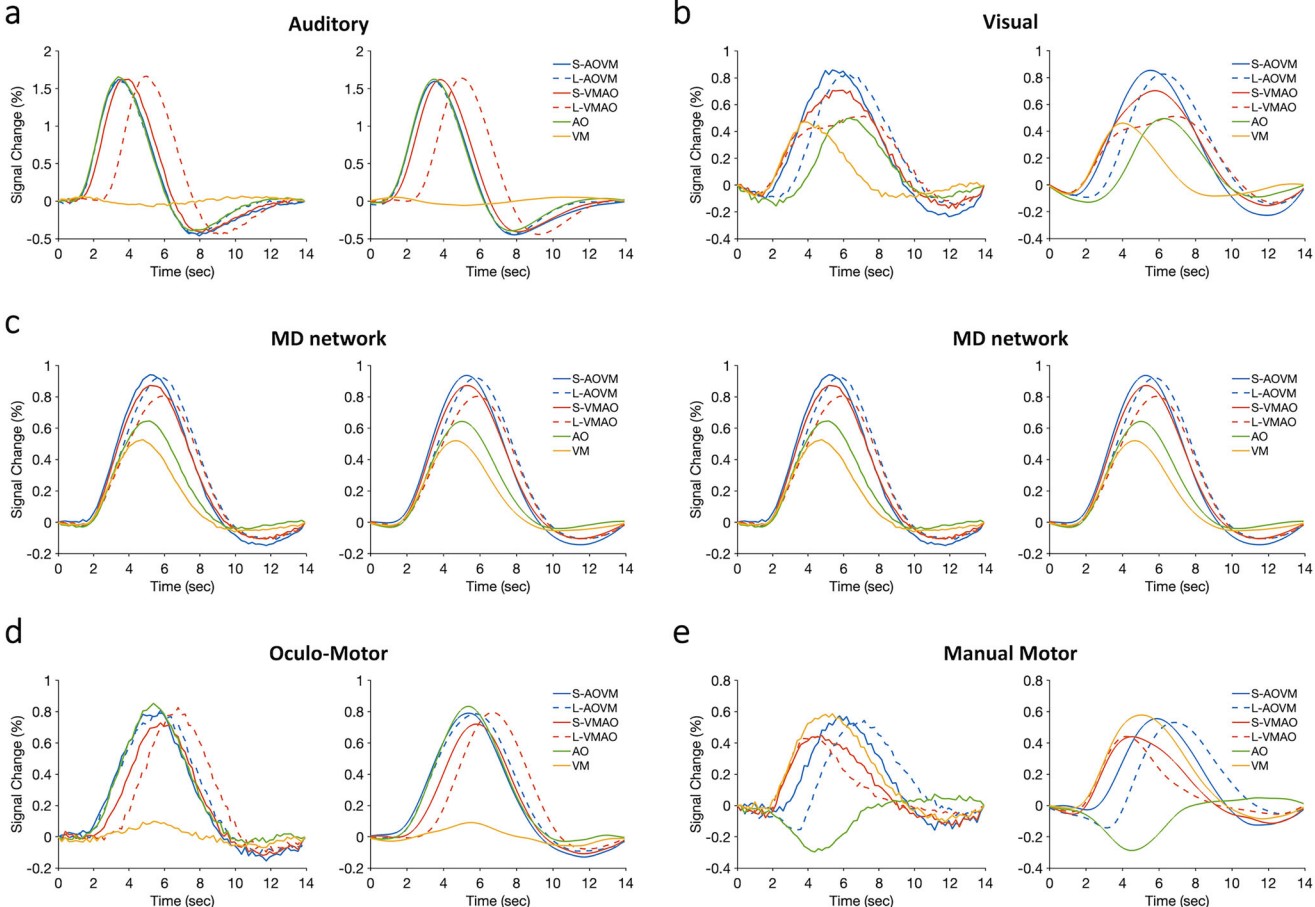

**Fig. 3 | BOLD univariate activation timecourses in single- and dual-task trials.**
Group-averaged timecourses of % signal change from the onset of T1 (0 ms) in
short-SOA AOVM (solid blue), long-SOA AOVM (dashed blue), short-SOA VMAO
(solid red), long-SOA VMAO (dashed red), single AO (green) and single VM (yellow)
trials in (**a**) auditory cortex, (**b**) visual cortex (V1–V4), (**c**) multiple-demand (MD)
cortical network, (**d**) oculomotor cortex (FEF), and (**e**) manual motor cortex (M1).

The left columns show the raw results and the right columns show the fitted curves
of the data. FEF frontal eye field, M1 primary motor cortex, SOA stimulus-onset
asynchrony, AO auditory-oculomotor, VM visual-manual. Note that the MD network
timecourses (**c**) have been duplicated on the left and right sides of the figure to
facilitate comparison of their activation profiles to the sensory and motor areas of
the AO and VM tasks. Source data are provided as a Source Data file.

results presented in Fig. 3c for the combined MD network areas—see
Supplementary Fig. 7 for individual MD network ROIs—are consistent
with these predictions; there were negligible onset latency differences
and pronounced peak latency differences (Onset latencies: short-SOA
dual-task vs. single-task conditions, 2.031 s for S-AOVM, 2.184 s for
S-VMAO vs. 2.192 s for single-AO, and 1.955 s for single-VM,
$F(3,75) = 1.54$, $p = 0.21$, partial $\eta^2 = 0.06$; the small L-SOA dual-task vs.
single-task difference ($F(3,75) = 5.76$, $p = 0.007$, partial $\eta^2 = 0.19$) can be
accounted for by the fact that large peak latency differences lead to
small onset latency differences, see simulations in Supplementary
Figs. 6 and 11. Peak activation latencies—a reliable measure of the
duration of activity[40]—in dual-task relative to single-task conditions:
5.272 s for S-AOVM, 5.306 s for S-VMAO, 5.793 s for L-AOVM and 5.931 s
for L-VMAO, vs. 5.035 s for single-AO and 4.720 s for single-VM,
$ps < 1.1 \times 10^{-4}$, Cohen's $ds > 0.89$). In addition, dual-task conditions
show greater activation amplitude than single-task conditions
($ps < 6.1 \times 10^{-8}$, Cohen's $ds > 1.48$). No other cortical or sub-cortical
areas showed this temporal BOLD response profile (see Supplemen-
tary Fig. 4).

Finally, the motor response ROIs should exhibit a temporal profile
of activity that is distinct from both sensory and MD network ROIs, as
their timecourses should be tracking the response times rather than
the timing of sensory stimulus presentations or central processing
times. The results are consistent with these predictions. In the manual
motor ROI, BOLD activity onsets at the same time in the single-task and

dual-task conditions irrespective of SOA as long as the VM task is the
first one executed ($F(2,50) = 0.61$, $p = 0.55$; Onset latency = 2.196 s for
S-VMAO vs. 1.992 s for single-VM, $p = 0.34$; Onset latency = 2.045 s for
L-VMAO vs. single-VM, $p = 0.75$) (Fig. 3e). However, when the VM task is
processed second in the dual-task conditions, the manual ROI's time-
course is shifted to the right, and more so at the Long SOA than at the
short SOA (Onset latency = 2.785 s for S-AOVM vs. 4.318 s for L-AOVM,
$t(25) = 5.77$, $p = 5.2 \times 10^{-6}$, Cohen's $d = 1.13$), owing to the fact that the
sum of the response times for the VM task and of the short 300 ms
AOVM SOA (T2 RT = 1.662 s in S-AOVM) is shorter than the sum of the
response times for the VM task and of the long AOVM 1500 ms SOA (T2
RT = 1.12 s in L-AOVM; 1.662 s + 0.3 s of short SOA vs. 1.12 s + 1.5 s of
long SOA, $t(25) = 26.04$, $p = 1.2 \times 10^{-19}$, Cohen's $d = 5.1$). It is not as
straightforward, however, to apply the same statistical comparison
between the dual-task and the single-task conditions because the
manual ROI activity is suppressed during execution of the AO task (see
Fig. 3e), which would equally affect the two dual-task timecourses but
not the single VM task timecourse. Therefore, we simply note here
that, as expected, the dual-task timecourses are visibly delayed relative
to the single-task VM timecourse (Fig. 3e).

The temporal profile of activation of the oculomotor ROI is similar
to that of the manual ROI except that here the FEF is not suppressed
during the visuo-manual task but is instead slightly activated (Fig. 3d).
This activation is likely related to the presentation of the visual sti-
mulus in the VM condition, as the FEF is well known to respond to

visual stimulation[41,42]. When the AO task was presented second, the dual-task conditions are delayed relative to the single-task AO condition (Fig. 3d; Onset latency = 3.306 s for L-VMAO vs. single-AO, $t(25) = 7.26$, $p = 1.3 \times 10^{-7}$, Cohen's $d = 1.42$; Peak latency = 6.771 s for L-VMAO vs. single-AO, $t(25) = 17.50$, $p = 1.5 \times 10^{-15}$, Cohen's $d = 3.43$; Onset latency = 2.177 s for S-VMAO vs. 2.117 s for single-AO, $p = 0.77$; Peak latency = 5.687 s for S-VMAO vs. 5.396 s for single-AO, $t(25) = 2.18$, $p = 0.04$, Cohen's $d = 0.43$), with the long SOA condition initiating and peaking later than the short-SOA condition (Onset latency: L-VMAO vs. S-VMAO, $t(25) = 5.33$, $p = 1.6 \times 10^{-5}$, Cohen's $d = 1.04$; Peak latency: L-VMAO vs. S-VMAO, $t(25) = 7.68$, $p = 4.9 \times 10^{-8}$, Cohen's $d = 1.5$). No such differences were observed when the AO task was presented first (Onset latency: $F(2,50) = 1.07$, $p = 0.35$; 1.912 s for S-AOVM vs. 2.117 s for single-AO, $p = 0.16$; 2.122 s for L-AOVM vs. single-AO, $p = 0.97$; Peak latency: $F(2,50) = 1.53$, $p = 0.23$; 5.266 s for S-AOVM vs. 5.396 s for single-AO, $p = 0.15$; 5.489 s for L-AOVM vs. single-AO, $p = 0.56$).

Taken together, the univariate SPM and timecourse analyses are consistent with a dissociation between sensory, central amodal, and motor response stages of neural information processing, as postulated by classic psychological models of cognitive information processing (e.g., refs. 5,6).

## Multivariate analyses

The univariate analyses are limited in that they do not provide direct evidence for serial queuing of information processing, as they cannot distinguish task-specific activity. To address this issue, we turned to a multi-voxel pattern analysis (MVPA; see ref. 43). Specifically, independently for each single sensorimotor task, we trained a classifier to distinguish between each of the 8 sensory-motor activity patterns in each sensory, motor, and MD network ROI. This methodological approach should result in two pattern classifiers; each uniquely tuned to one of the two tasks (see "Methods"), thus allowing us to track task-specific activity coursing throughout the ROIs. Below, we first present the results for single-task trials followed by the results for the dual-task trials.

**Single-task trials decoding.** With the help of a fourth GLM to estimate the regression coefficients (beta values) for each individual trial of each S-R mapping, we independently trained the two pattern classifiers to discriminate the trials of one S-R mapping against the trials of the other seven S-R mappings, and tested these trained classifiers on their respective task trials using a leave-one-out cross validation procedure (see "Methods"). First turning to the auditory cortex ROI (Fig. 4a), the AO pattern classifier after training was well above chance (12.5%) at classifying AO trials ($p = 1.29 \times 10^{-9}$, FDR corrected for multiple comparisons, ref. 44) but not VM trials ($p = 0.62$, FDR corrected). By contrast, the VM pattern classifier failed to successfully classify either AO ($p = 0.69$, FDR corrected) or VM trials ($p = 0.45$, FDR corrected) in the auditory cortex ROI. These multivariate results are consistent with the univariate results in highlighting the sensory-specific nature of information processing in the auditory cortex. The pattern classification results in the visual cortex ROI also reflect the univariate findings (Fig. 4b): Not only is the VM pattern classifier able to distinguish between VM trials ($p = 3.71 \times 10^{-5}$, FDR corrected), but the AO classifier is even better at distinguishing between AO trials ($p = 5.18 \times 10^{-5}$, FDR corrected), consistent with the massive visual cortex activation in the AO task (see Figs. 2 and 3). These univariate and multivariate results severely compromise the interpretability of the findings in the visual ROIs in dual-task conditions (see Supplementary Fig. 8). Hence, the dual-task findings presented below will focus on the auditory cortex as a barometer of activity in a purely sensory ROI.

The manual motor cortex ROI showed a decoding pattern expected of a modality-specific motor area (Fig. 4d). Specifically, the pattern classified trained on VM tasks could distinguish between VM trials ($p = 4.18 \times 10^{-11}$, FDR corrected) but not between AO trials ($p = 0.34$, FDR corrected) in that ROI. By contrast, the AO pattern classifier failed to decode either AO ($p = 0.08$, FDR corrected) or VM trials ($p = 0.98$, FDR corrected). Thus, the multivariate results are consistent with the univariate results in demonstrating modality-specific activity during single-task conditions in the primary motor cortex.

Decoding in the other motor ROI, the FEF, had generally poor signal-to-noise ratio (SNR) under all single-task conditions (Fig. 4c). Nevertheless, the pattern classifier trained on AO tasks could distinguish between AO trials ($p = 0.006$, FDR corrected) but not between VM trials ($p = 0.55$, FDR corrected), and the VM pattern classifier failed to decode either VM ($p = 0.1$, FDR corrected) or AO trials ($p = 0.26$, FDR corrected). Even though the single-task decoding results in the FEF are as expected of a motor ROI, the poor SNR precluded any significant conclusions to be drawn in the Dual-task conditions (see Supplementary Fig. 9). For this reason, the dual-task decoding results for motor cortex presented further below will focus on the manual ROI.

Finally, if the central MD network ROIs process both sensori-motor tasks, they should show a distinct decoding patten from both the sensory and motor ROIs. Specifically, the AO pattern classifier should distinguish between AO trials but not VM trials, and conversely the VM classifier should discern between the VM trials but not the AO trials. The results bore out these predictions (Fig. 5): AO decoding between AO trials, $p = 2.06 \times 10^{-5}$, FDR corrected; AO decoding between VM trials, $p = 0.56$, FDR corrected; VM decoding between AO trials, $p = 0.56$, FDR corrected; VM decoding between VM trials, $p = 2.34 \times 10^{-5}$, FDR corrected.

In sum, multivariate analyses of single-task trial conditions revealed the feasibility of using pattern classifiers to distinguish task-specific activity in sensory (auditory), central and motor (M1) cortical areas throughout the various stages of information processing.

**Dual-task trials decoding.** Tracking the activity of each task as it proceeds through the sensory, central and motor ROIs during dual-task performance requires not only that each pattern classifier's decoding performance be specific to one of the two tasks, but that it can also capture the dynamic changes in its decoding accuracy throughout the course of a trial. We achieved this by testing the classifiers trained above against the beta estimates of *each time point* of each trial obtained from a fifth GLM analysis (see "Methods").

Using the auditory cortex as modality-specific sensory ROI, Fig. 6a shows the timecourse of decoding under both single-task and dual-task conditions at short and long SOA, for both averaged raw and curve-fit data, with the dual-task timecourses back-shifted by their SOAs in order to compare the timecourses under single- and dual-task conditions. As shown in Fig. 6a, AO task decoding timecourses are indistinguishable under single and both dual-task conditions at long SOA as one would predict if the SOA is long enough to avoid dual-task interference (peak latency: $F(2,50) = 0.71$, $p = 0.5$; L-AOVM vs. single-AO, $p = 0.67$, L-VMAO vs. single-AO, $p = 0.67$, L-AOVM vs. L-VMAO, $p = 0.73$, FDR corrected; onset latency: $F(2,50) = 1.7$, $p = 0.19$; L-AOVM vs. single-AO, $p = 0.67$, L-VMAO vs. single-AO, $p = 0.2$, L-AOVM vs. L-VMAO, $p = 0.2$, FDR corrected). At short SOA, similar results were observed for the AO single-task and dual-task conditions when AO was the second of the two tasks (peak latency: S-VMAO vs. single-AO, $p = 0.12$; onset latency: $p = 0.18$). However, decoding began earlier—though it persisted just as long—when AO was the first of the two tasks (peak latency: S-AOVM vs. single-AO, $p = 0.0013$; onset latency: $p = 0.012$), perhaps as part of a process to attenuate dual-task interference (see "Discussion"). Asides from this early onset of decoding for Task1 at short SOA, the results in the auditory cortex suggest that the timecourse of sensory information processing is largely unaffected under dual-task conditions relative to single-task conditions.

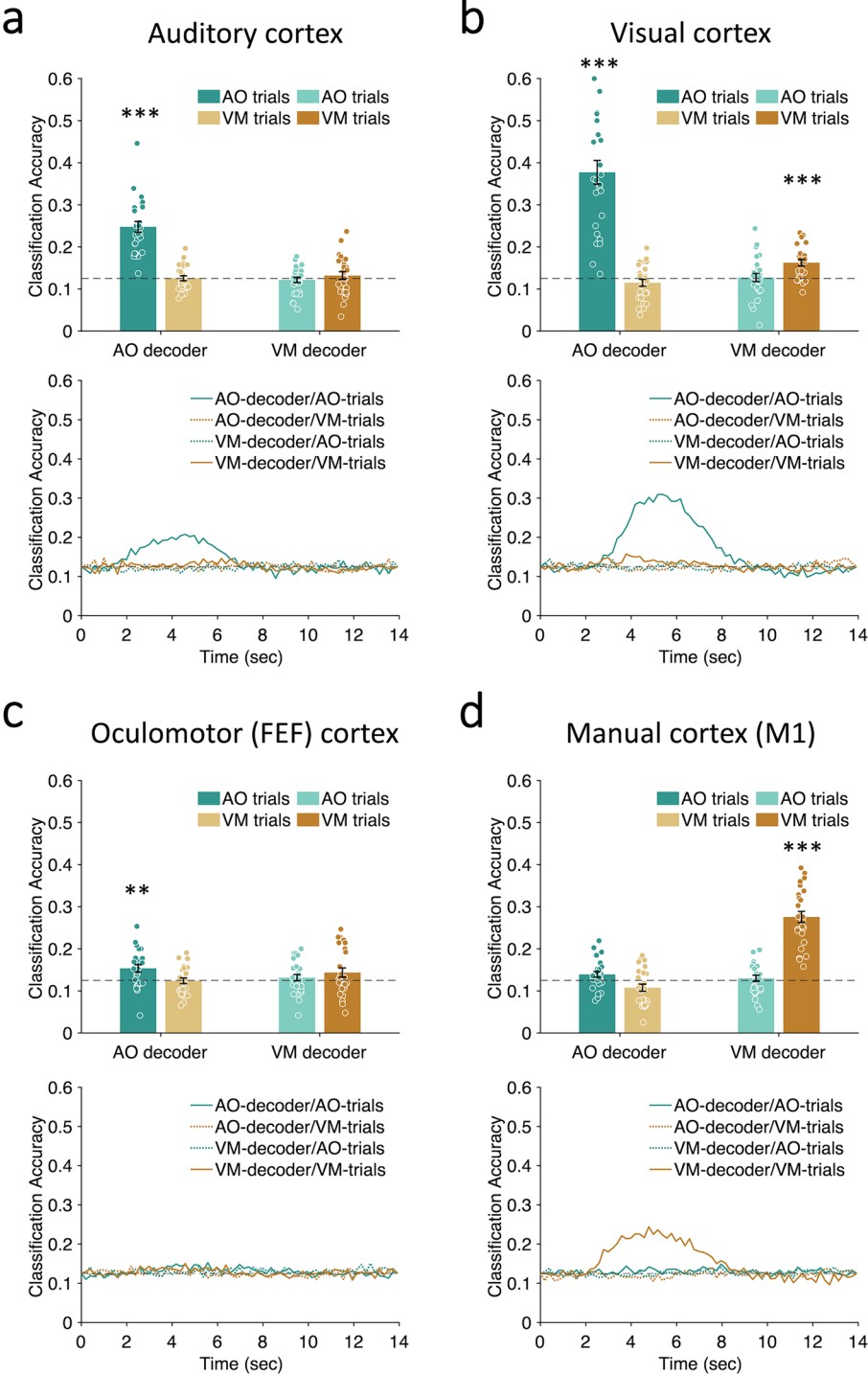

**Fig. 4 | Single-task multivariate decoding in sensory and motor ROIs. a** Auditory cortex; (**b**) visual cortex; (**c**) oculomotor cortex; and (**d**) manual motor cortex. The upper graph in each panel shows the group-averaged classification accuracy in the single task trials with color-filled dots representing individuals' data. The *x*-axis represents task-specific classifiers and colors represent different single-task trial types. The darker colors represent the congruous decoder-trial type pairs whereas the lighter colors represent the incongruent pairs. **a** Auditory cortex (AO decoding between AO trials, $p = 1.29 \times 10^{-9}$); (**b**) visual cortex (VM decoding between VM trials, $p = 3.71 \times 10^{-5}$; AO decoding between AO trials, $p = 5.18 \times 10^{-5}$); (**c**) oculomotor cortex (AO decoding between AO trials, $p = 0.006$); (**d**) manual motor cortex (VM decoding between VM trials, $p = 4.18 \times 10^{-11}$). All statistical tests in **a**–**d** are one-sample *t*-tests, one-tailed, FDR corrected, $N = 26$. The lower graph in each panel shows the average classification accuracy time series in the single tasks, with the solid lines representing task-specific decoding for the congruous trial type and dashed lines representing results for the incongruous trial type. Dashed gray lines represent chance levels (12.5%). Error bars represent the standard error of the mean. Asterisks indicate the significance of the decoding against the chance level; **$p < 0.01$, ***$p < 0.001$. AO auditory-oculomotor, VM visual-manual. Source data are provided as a Source Data file.

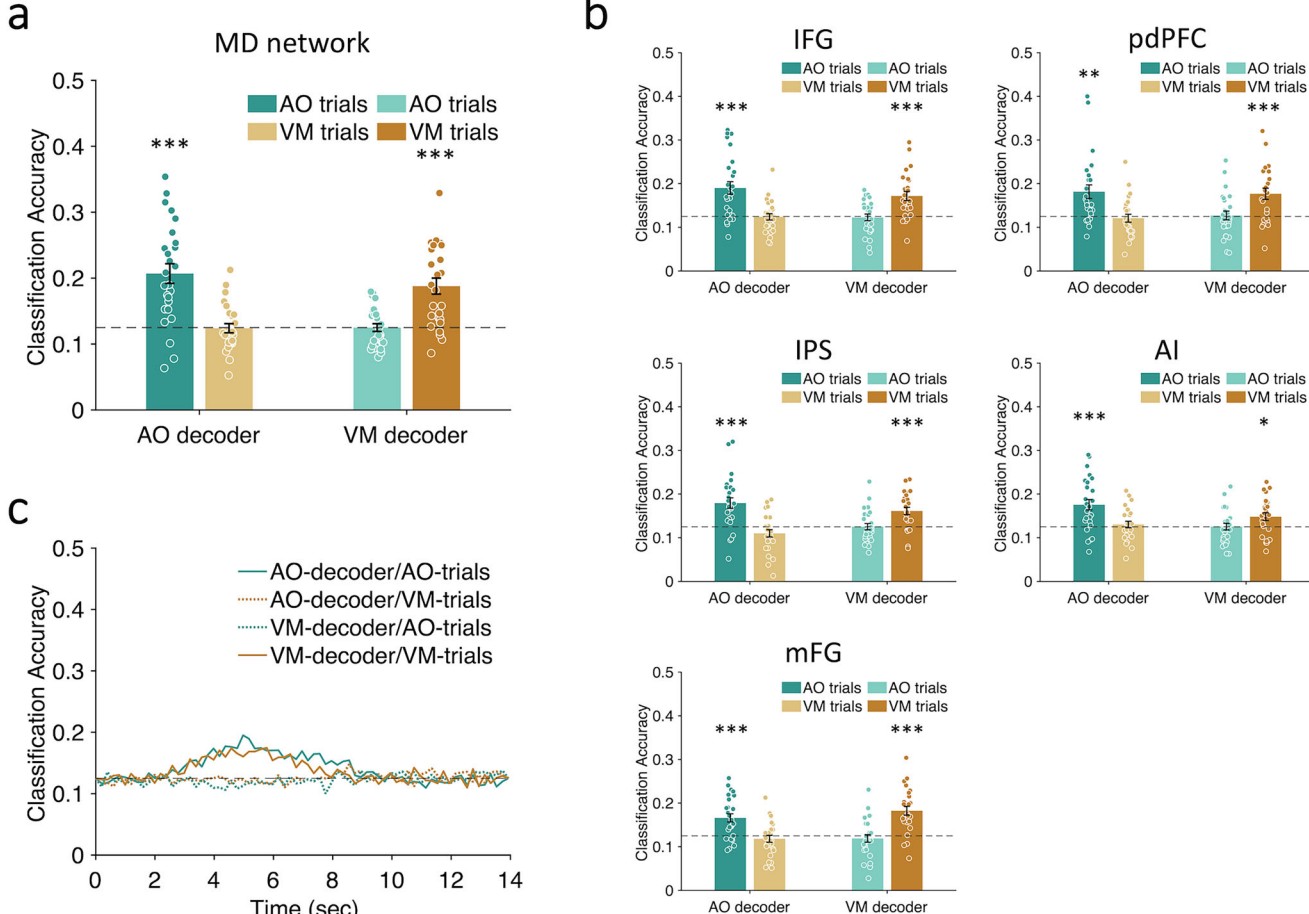

**Fig. 5 | Single-task decoding in the Multiple-Demand (MD) network.** The average classification accuracies in the single-task trials in (**a**) MD network (AO decoding between AO trials, $p = 2.06 \times 10^{-5}$; VM decoding between VM trials, $p = 2.34 \times 10^{-5}$) and (**b**) individual bilateral ROIs of the MD network (in IFG, AO decoding between AO trials, $p = 1.24 \times 10^{-4}$, VM decoding between VM trials, $p = 1.24 \times 10^{-4}$; in pdPFC, AO decoding between AO trials, $p = 0.0013$, VM decoding between VM trials, $p = 8.55 \times 10^{-4}$; in IPS, AO decoding between AO trials, $p = 1.67 \times 10^{-4}$, VM decoding between VM trials, $p = 1.67 \times 10^{-4}$; in AI, AO decoding between AO trials, $p = 5.18 \times 10^{-4}$, VM decoding between VM trials, $p = 0.015$; in mFG/dACC, AO decoding between AO trials, $p = 1.69 \times 10^{-4}$, VM decoding between VM trials, $p = 1.21 \times 10^{-5}$). All statistical tests in (**a**, **b**) are one-sample $t$-tests, one-tailed, FDR corrected, $N = 26$. Color-filled dots represent individuals' data. **c** Group-averaged classification accuracies time series in the single task trials in the MD network. Labels and axes are as described in Fig. 4 legend. IFG inferior frontal gyri, pdPFC posterior dorsal prefrontal cortex, IPS intraparietal sulcus, AI anterior insula, mFG/dACC medial frontal gyri/dorsal anterior cingulate cortex. Error bars represent the standard error of the mean. Asterisks indicate the significance of the decoding against chance level; *$p < 0.05$, **$p < 0.01$, ***$p < 0.001$. AO auditory-oculomotor, VM visual-manual. Source data are provided as a Source Data file.

In an analogous manner to the auditory cortex for sensory ROIs, we used the manual motor ROI to ascertain how task-specific information is processed in motor cortex under dual-task conditions, as that ROI showed robust modality-specific univariate activity and multivariate decoding in single-task conditions. At long SOA, the decoding timecourses are indistinguishable across single-task and dual-task conditions (Fig. 6b; peak latency: $F(2,50) = 0.95$, $p = 0.37$; L-VMAO vs. single-VM, $p = 0.49$, L-AOVM vs. single-VM, $p = 0.38$, L-VMAO vs. L-AOVM, $p = 0.93$, FDR corrected; onset latency: $F(2,50) = 1.56$, $p = 0.23$; L-VMAO vs. single-VM, $p = 0.86$, L-AOVM vs. single-VM, $p = 0.09$, L-VMAO vs. L-AOVM, $p = 0.29$, FDR corrected). At short SOA with the VM task presented first, peak decoding latency was similar to when that task was performed alone (peak latency: S-VMAO vs. single-VM, $p = 0.65$). By contrast, the entire decoding timecourse was postponed when the VM task was carried out second (peak latency: difference between S-AOVM and single-VM, $t(25) = 4.44$, $p = 0.00016$. Cohen's $d = 0.87$; onset latency difference between S-AOVM and single-VM, $t(25) = 4.72$, $p = 7.8 \times 10^{-5}$, Cohen's $d = 0.93$), as would be expected of a motor cortex area that processes a motor response that is delayed by the duration of the PRP (Fig. 1e).

Finally, we assessed the timecourses of decoding in the MD network ROIs to determine whether they show evidence of serial queuing of task processing as postulated by the central bottleneck model, collapsing across task order to improve SNR (see "Methods"). As illustrated in Fig. 7a (bottom panels), the temporal dynamics of Task1 and Task2 decoding were not different from the single-task timecourses at long SOA (peak latency: $F(2,50) = 0.13$, $p = 0.85$; L-T1 vs. single, $p = 0.77$, L-T2 vs. single-task, $p = 0.85$, L-T1 vs. L-T2, $p = 0.85$, FDR corrected; onset latency: $F(2,50) = 0.2$, $p = 0.82$; L-T1 vs. single-task, $p = 0.92$, L-T2 vs. single-task, $p = 0.92$, L-T1 vs. L-T2, $p = 0.92$, FDR corrected). This is as expected if there is no dual-task interference. By contrast, a postponement in Task2 decoding was clearly evidenced at short SOA (Fig. 7a, top panels; peak latency for S-T2 vs. single-task, $t(25) = 5.90$, $p = 3.7 \times 10^{-6}$, Cohen's $d = 1.16$; onset latency for S-T2 vs. single-task, $t(25) = 4.63$, $p = 9.79 \times 10^{-5}$, Cohen's $d = 0.91$), whereas Task1 decoding occurred along the same time frame as under single-task conditions (peak latency: S-T1 vs. single-task, $p = 0.86$; onset latency: S-T1 vs. single-task, $p = 0.83$). Noteworthily, the mean decoding peak latency difference between S-T2 and single-task in Fig. 7a (697 ms) bears resemblance to the magnitude of the PRP (cf. Fig. 1). If the congruency

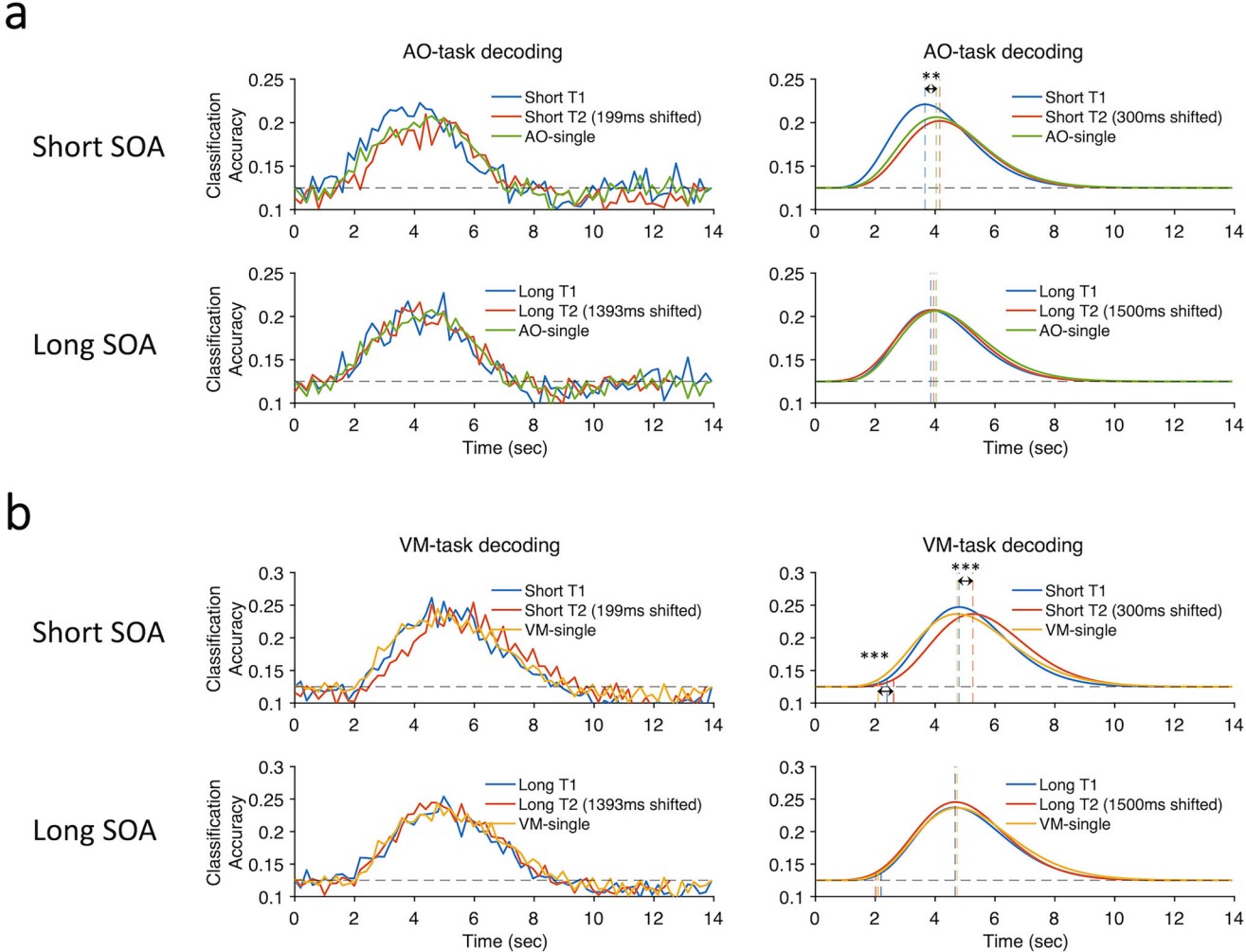

**Fig. 6 | Time courses of single-task and dual-task decoding in sensory and motor areas.** Group-averaged classification accuracy time series in the single-task and dual-task trials in (**a**) auditory cortex and (**b**) manual motor cortex. The upper graphs in each panel show the results for the short SOA conditions and the lower graphs for the long SOA conditions. The left columns show the raw decoding results and the right columns show the fitted curves of the decoding data. Blue lines show task-specific decoding results for T1 in the dual-task conditions and red lines show task-specific decoding results for T2 in the dual-task conditions. Green lines show AO-specific decoding for the single AO task in (**a**) auditory cortex and yellow lines show VM-specific decoding for the single VM task in (**b**) manual motor cortex. Dashed gray lines represent chance levels (12.5%). Vertical dashed lines indicate onset/peak latencies in single-task and dual-task trials. Asterisks indicate

significantly earlier T1 decoding peak latency in the short-SOA dual-task trials (marked by the blue vertical line) relative to the single-task trials (marked by the green vertical line) in (**a**) auditory cortex (\*\*$p = 0.0013$, $N = 26$, paired-sample $t$-test, two-tailed), and significantly postponed T2 decoding peak/onset latency in the short-SOA dual-task trials (marked by the red vertical line) relative to the single-task trials (marked by the yellow vertical line) in (**b**) manual-motor cortex (peak latency, \*\*\*$p = 1.6 \times 10^{-4}$, $N = 26$, paired-sample $t$-test, two-tailed; onset latency, \*\*\*$p = 7.8 \times 10^{-5}$, $N = 26$, paired-sample $t$-test, two-tailed). Note that the raw time-courses in the left panels could only be shifted by multiples of TRs to the nearest SOAs (199 ms and 1393 ms) whereas the curve-fitted data could be shifted by the exact SOAs (300 ms and 1500 ms). SOA stimulus-onset asynchrony, AO auditory-oculomotor, VM visual-manual. Source data are provided as a Source Data file.

between the group-averaged T2 decoding latency and magnitude of the PRP is more than coincidental, we might expect individual subjects' differences in T2 decoding latency to correlate with their PRP effect. Figure 7b shows that it is indeed the case. Specifically, individual subjects' differences in PRP magnitude correlated with their T2 decoding onset latencies at short ($r = 0.44$, $p = 0.026$) but not at long SOA ($r = 0.17$, $p = 0.41$). Their PRP magnitude also did not correlate with their T2 peak latency at either short ($r = 0.13$, $p = 0.53$) or long SOA ($r = 0.31$, $p = 0.13$), but that is not unexpected since individual differences in peak T2 latencies would not only be affected by the magnitude of the postponement caused by T1 processing but also by individual differences in processing Task2, which is not taken into account in the PRP magnitude. Taken together, these results not only reveal neural evidence for serial queuing of task-specific activity in the MD network, they also suggest that this serial queuing may account for the behavioral PRP effect.

## Chronometric flow of information processing across sensory, central and motor ROIs

The results of the time-resolved MVPAs describe the temporal profiles of task-specific decoding within sensory, central, and motor cortical areas. Based on these profiles and on classic models[6,45–47], we surmised that information processing in the central bottleneck should take place at a different latency than sensory and motor stages of information processing. Specifically, perceptual processing in sensory areas should occur prior to the central bottleneck of information processing in the MD network, whereas motor response processing should take place after central processing. To test these assumptions, we compared the chronometry of decoding across the sensory, central and motor ROIs. If there is a temporal gradient of information processing from sensation to response selection to motor execution, decoding should peak earlier in sensory than in MD network areas, and decoding in the MD network areas should in turn peak earlier than in the motor response

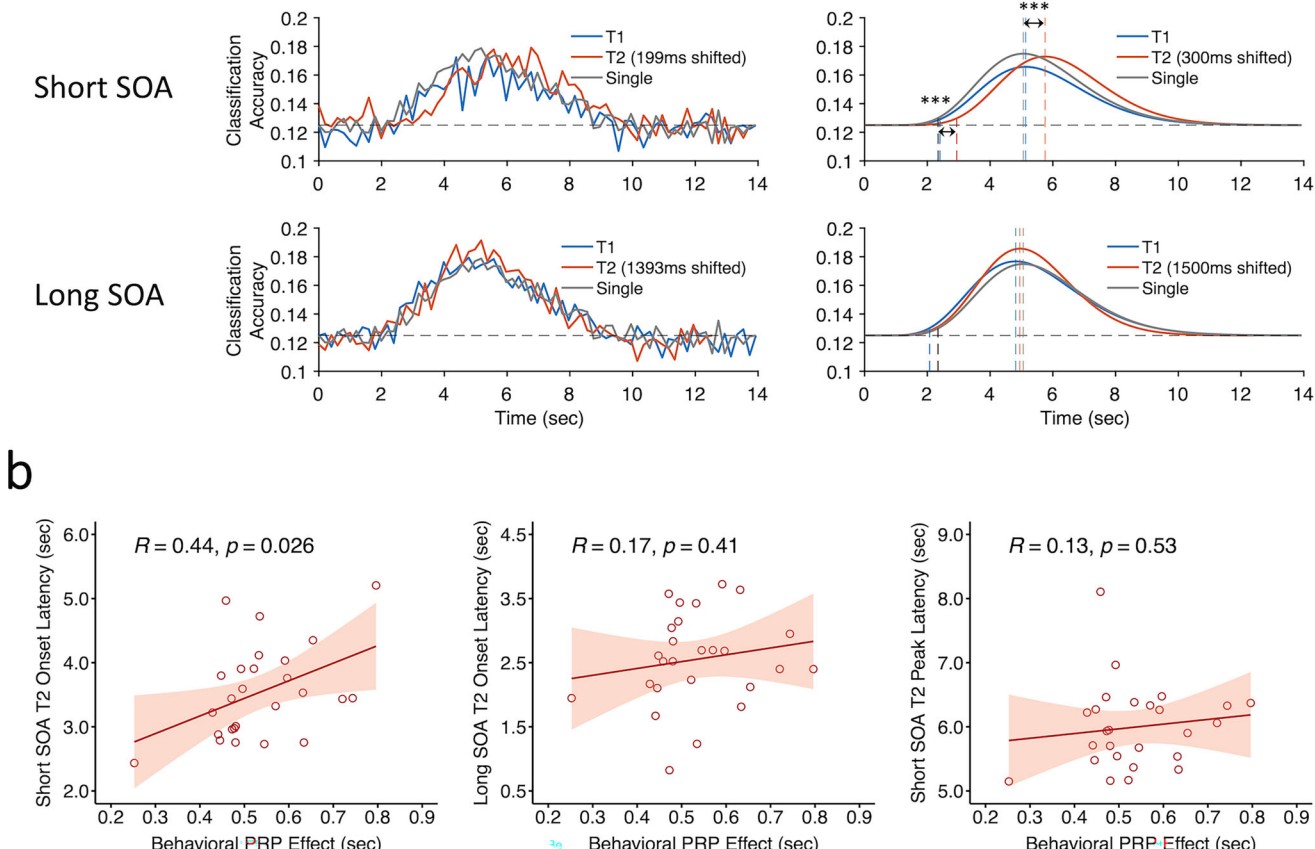

**Fig. 7 | Time courses of single-task and dual-task decoding in the MD network. a** The group-averaged classification accuracy time series in the MD network. The upper graphs show results of task-specific decoding for the short-SOA dual-task trials and the lower graphs for the long-SOA dual-task trials. The left column shows the raw decoding results and the right column shows the fitted curves of the decoding results. Dashed gray lines represent the chance levels (12.5%) and the colored vertical dashed lines indicate decoding peak latencies and onset latencies. **b** Correlations between individuals' fMRI decoding measures (peak latency and onset latency) and magnitude of the behavioral PRP (RT2 difference between short- and long-SOA dual-task conditions). Hollow dots represent individual data. The red line represents a linear fit and the shaded ribbon represents the standard error.

Asterisks indicate significantly postponed T2 decoding peak latencies and onset latencies in the short-SOA dual-task trials (marked by the red vertical line) relative to the single-task trials (marked by the gray vertical line): peak latency, ***$p = 3.7 \times 10^{-6}$, $N = 26$, paired-sample $t$-test, two-tailed; onset latency, ***$p = 9.79 \times 10^{-5}$, $N = 26$, paired-sample $t$-test, two-tailed. Note that the raw time-courses in the left panels could only be shifted by multiples of TRs to the nearest SOAs (199 ms and 1393 ms) whereas the curve-fitted data could be shifted by the exact SOAs (300 ms and 1500 ms). MD multiple-demand, SOA stimulus-onset asynchrony, PRP psychological refractory period. Source data are provided as a Source Data file.

ROIs. Such gradient should be particularly evident at peak latency as the latter includes the extended processing duration that takes place at the stage of response selection, which for 8AD tasks should make up the bulk of the 1 s response times compared to purely sensory or response execution processing.

Figure 8 compares the decoding timecourses across sensory, MD network and motor ROIs in the AO and VM single tasks. As predicted, decoding peaked earlier in the sensory areas than in the MD network and motor areas (For the single-AO task: auditory sensory vs. MD network, $t(25) = 5.42$, $p = 1.26 \times 10^{-5}$, Cohen's $d = 1.06$; auditory sensory vs. oculomotor, $t(25) = 3.54$, $p = 0.0016$, Cohen's $d = 0.69$. For the single-VM task: visual sensory vs. MD network, $t(25) = 3.26$, $p = 0.0032$, Cohen's $d = 0.64$; visual sensory vs. manual motor, $t(25) = 4.08$, $p = 0.0004$, Cohen's $d = 0.8$). However, there were no such peak latency delay between the MD network and motor ROIs (For the single-AO task: MD network vs. oculomotor, $p = 0.9$. For the single-VM task: MD network vs. manual motor, $p = 0.46$). This finding is inconsistent with the interpretation that the motor ROIs correspond to the stage of response execution that follows the stage of response selection. We considered whether these results could simply be due to intrinsic

differences in hemodynamic properties across these brain regions. That is to say, the MD network and motor ROIs may simply be equally more sluggish in their hemodynamic responses relative to the sensory ROIs. However, as we shall see with the Quartile RT analyses further below (Fig. 9), the timecourses of activations of the sensory, central and motor ROIs are differentially sensitive to cognitive processing times, ruling out a simple intrinsic-differences-in-hemodynamic-response account for the present peak activation latency results. An alternative explanation for the similar temporal profiles of decoding in the MD network and motor ROIs is that the latter do not exclusively act at the motor end-stage of information processing but also participate in response selection. Indeed, much of the results in the motor ROIs presented so far are compatible with a response selection account of motor cortex activation (cf. Univariate BOLD results in Fig. 3d, e; single-task decoding in Fig. 4c, d and dual-task decoding in Fig. 6b).

**Single-task quartile RT analysis**
To address whether the motor ROIs are involved in the central stage of response selection, we took advantage of the well-known finding that

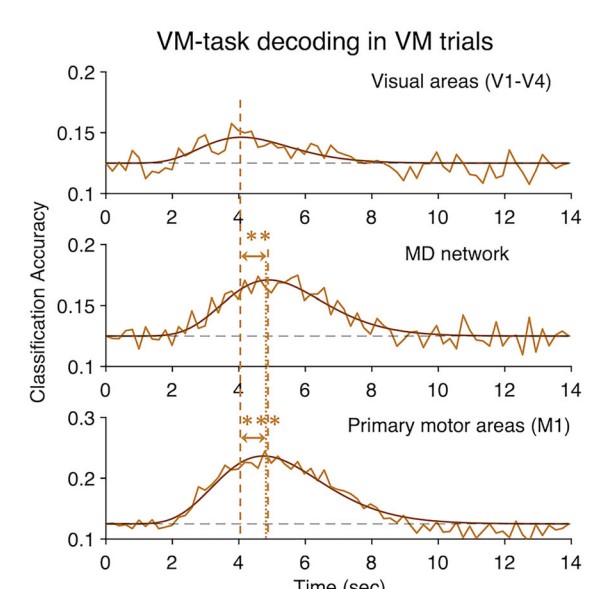

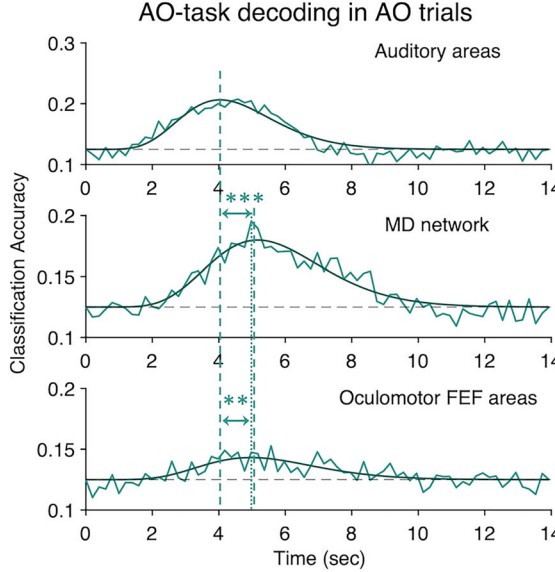

**Fig. 8 | Comparative decoding time courses in single VM and AO tasks.** Chronometry of decoding peak latency across sensory, MD network and motor areas in (**a**) single VM task and (**b**) single AO task. While the decoding timecourses peaked earlier in the sensory than in the MD network (single VM task, $p = 0.0032$; single AO task, $p = 1.26 \times 10^{-5}$, $N = 26$, paired-sample $t$-tests, two-tailed) or motor areas (single VM task, $p = 0.0004$; single AO task, $p = 0.0016$, $N = 26$, paired-sample $t$-tests, two-tailed), the latter two showed no peak latency differences ($p$'s > 0.4).

Each subplot shows superimposed raw and curve-fitted decoding time series. Asterisks indicate significantly different decoding peak latencies between the sensory (marked by the left dashed vertical line) and the MD network areas (marked by the right dashed vertical line) and between the sensory and motor areas (marked by the dotted vertical line), \*\*$p < 0.01$, \*\*\*$p < 0.001$. MD multiple-demand, AO auditory-oculomotor, VM visual-manual. Source data are provided as a Source Data file.

variability in response times primarily originates at the stage of response selection[48–50], at least when data information is not limited and stimuli and responses are clearly distinct from one another, as were the cases in the present study. The response selection-related variability should be particularly pronounced in our 8 response-mapping tasks, as the overall duration of response selection scale with the number of alternatives[50,51] and RT variability is proportional to response times[52]. Consistent with these predictions, examination of individuals' RT distributions revealed considerable variability in their response times (Supplementary Fig. 10). We binned these response times into quartiles and tested the hypothesis that brain regions involved in response selection should show a proportional increase in univariate BOLD response peak latency (and amplitude) with longer RT quartiles while their onset latencies should remain unaffected. If, on the other hand, the motor ROIs are solely involved in response execution downstream from the stage of response selection, they should only show onset latency delays as their activation timecourses are simply postponed with longer quartile RTs.

Figure 9 shows the results of this quartile RT analysis for the AO and VM single-task conditions in the sensory, MD network, and motor ROIs. The sensory cortical areas showed no onset or peak latency differences across quartiles (Fig. 9a, b; for single-VM: onset latency, $F(3,75) = 0.99$, $p = 0.4$, partial $\eta^2 = 0.04$; peak latency, $F(3,75) = 1.83$, $p = 0.15$, partial $\eta^2 = 0.07$; for single-AO: onset latency, $F(3,75) = 0.1$, $p = 0.96$, partial $\eta^2 = 0.004$; peak latency, $F(3,75) = 2.63$, $p = 0.06$, partial $\eta^2 = 0.1$). This is consistent with the notion that these perceptual areas participate in pre-central stages of information processing. An individual differences analysis corroborated these group-averaged results: there were no correlations between individual subjects' RT differences between the 4th RT quartile (Q4) and the 1st RT quartile (Q1) and their onset or peak latency differences between Q4 and Q1 (Fig. 9a, b, lower panels; for single-VM: peak latency, $r = 0.039$, $p = 0.85$; onset latency, $r = -0.3$, $p = 0.14$; for single-AO: peak latency, $r = 0.064$, $p = 0.76$; onset latency, $r = 0.067$, $p = 0.75$).

In contrast to the sensory areas, the MD network ROIs showed clear peak latency (and amplitude) differences across quartiles (Fig. 9c, d; for single-VM: peak latency, $F(3,75) = 22.83$, $p = 1.33 \times 10^{-10}$, partial $\eta^2 = 0.48$; peak amplitude, $F(3,75) = 15.37$, $p = 1.22 \times 10^{-6}$, partial $\eta^2 = 0.38$; for single-AO, peak latency, $F(3,75) = 10.67$, $p = 6.4 \times 10^{-6}$, partial $\eta^2 = 0.30$; peak amplitude, $F(3,75) = 12.64$, $p = 1.46 \times 10^{-5}$, partial $\eta^2 = 0.34$). There were small onset latency differences accentuated here by an initial dip (For single-VM: $F(3,75) = 6.79$, $p = 0.0004$, partial $\eta^2 = 0.21$. For single-AO, $F(3,75) = 2.18$, $p = 0.1$, partial $\eta^2 = 0.08$), but that is not surprising since peak latency differences are expected to lead to small onset latency differences (Supplementary Fig. 11). Importantly, such onset differences cannot account for the observed peak amplitude differences across quartiles, nor can it explain the broadening of the hemodynamic response with increased RTs (full-width-at-half-maximum [FWHM] quartile effect: $F(3,75) = 2.75$, $p < 0.05$, partial $\eta^2 = 0.10$), whereas those two effects are predicted by increases in the duration of the neural event of response selection (Supplementary Fig. 11). Furthermore, these group-averaged results are supported by individual differences analyses, which show that participants' RT differences between Q4 and Q1 varied with their peak latencies (Fig. 9c, d; lower panels; for single-VM, $r = 0.69$, $p = 9.8 \times 10^{-5}$; for single-AO, $r = 0.5$, $p = 0.0087$), but not their onset latencies (for single-VM, $r = 0.13$, $p = 0.51$; for single-AO, $r = 0.076$, $p = 0.71$). In sum, the MD network areas behaved as expected of brain regions involved in response selection and provide a hemodynamic benchmark for evaluating the possible involvement of motor cortex ROIs in response selection.

As illustrated in Fig. 9e, f, the BOLD responses in motor cortex ROIs were just like those in the MD network ROIs. There were peak latency differences between the quartiles (single-VM: peak latency, $F(3,75) = 11.15$, $p = 3.93 \times 10^{-6}$, partial $\eta^2 = 0.31$; single-AO: peak latency, $F(3,75) = 7.69$, $p = 8.2 \times 10^{-4}$, partial $\eta^2 = 0.24$), but no onset latency differences (single-VM: $F(3,75) = 0.77$, $p = 0.5$, partial $\eta^2 = 0.03$; single-AO: $F(3,75) = 1.01$, $p = 0.39$, partial $\eta^2 = 0.04$). The individual differences

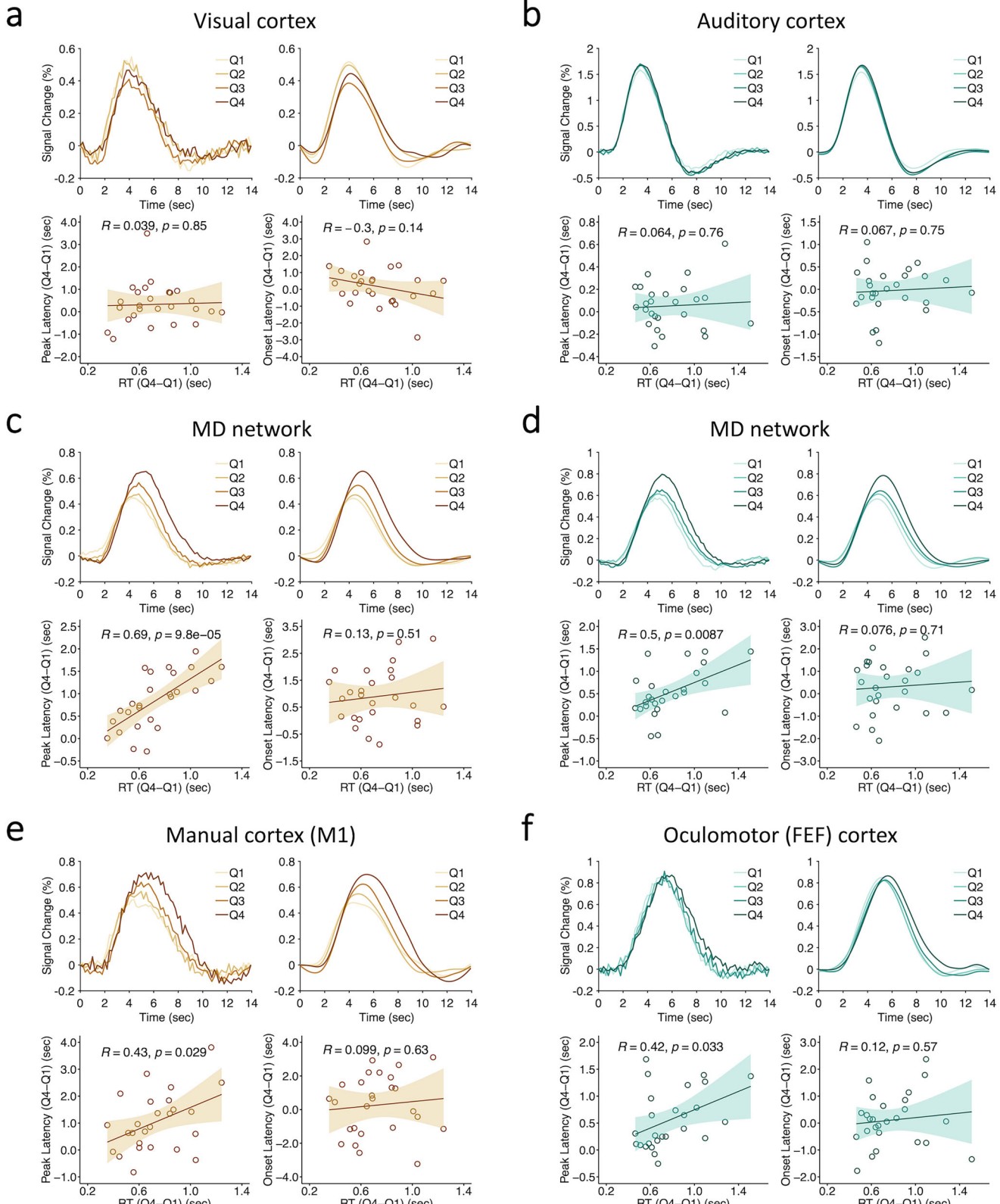

**Fig. 9 | Single-task quartile-RT analysis.** Single-task quartile-RT results in sensory (**a** visual; **b** auditory), (**c**, **d**) MD network, and motor (**e** manual; **f** oculomotor) areas for the VM (**a**, **c**, **e**) and AO (**b**, **d**, **f**) tasks. In each subplot, the upper graphs show the raw time series of BOLD signal changes (left) and their fitted curves (right) for the four quartile-RT bins. The lower graphs show individuals' correlations between their quartile peak latency (left) and onset latency (right) differences (Q4 vs. Q1)

and their quartile RT differences (Q4 vs. Q1). Hollow dots represent individuals' data. The solid line represents a linear fit and the shaded ribbon represents the standard error. MD multiple-demand, VM visual-manual, AO auditory-oculomotor, Q1 the first quartile, Q2 the second quartile, Q3 the third quartile, Q4 the fourth quartile. Source data are provided as a Source Data file.

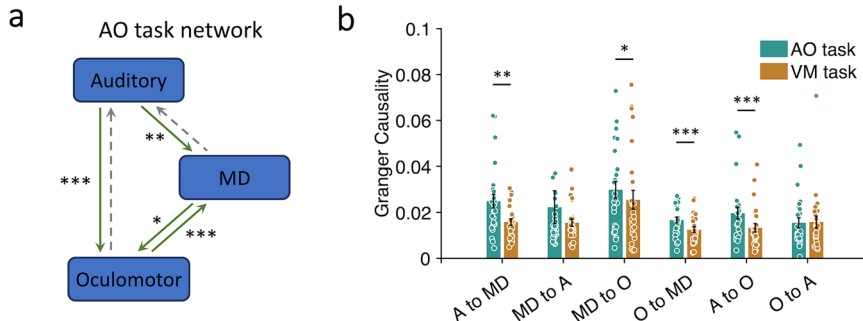

**Fig. 10 | Granger causality analysis of the AO network in single-task condition.**
**a** AO task neural network model. Green solid arrows indicate the direction of the significant causal influence of a brain area onto another tested in this analysis. Dashed arrows indicate that the direction of the causal influence is not significant. **b** Granger causality estimates for each of the pairwise directional connections during the AO and the VM tasks in the AO neural network (from A to MD, $p = 0.002$; from MD to O, $p = 0.02$; from O to MD, $p = 0.0005$; from A to O, $p = 0.0003$; All statistical tests are paired-sample $t$-tests, two-tailed, $N = 26$). A auditory areas, MD multiple-demand network, O oculomotor areas. Error bars represent the standard error of the mean. Asterisks represent statistically different Granger values between the AO and VM tasks; *$p < 0.05$, **$p < 0.01$, ***$p < 0.001$. AO auditory-oculomotor, VM visual-manual. Source data are provided as a Source Data file.

analyses backed up the group-averaged results: there was a correlation between participants' quartile RT differences and their peak latency quartile differences (Fig. 9e, f, lower panels; single-VM: $r = 0.43$, $p = 0.029$; single-AO: $r = 0.42$, $p = 0.033$) but not onset latency differences (single-VM: $r = 0.099$, $p = 0.63$; single-AO: $r = 0.12$, $p = 0.57$). These results clearly indicate that the motor cortex ROIs behave like regions involved in response selection rather than simply response execution. Unlike the MD network, however, these motor areas' participation in response selection is modality-specific: the manual motor cortex did not activate during response selection of the oculomotor task, and vice versa for the FEF with the manual task (Fig. 3). Importantly, the absence of peak latency differences in the sensory ROIs indicates that the response selection results in the MD network and motor cortex ROIs are not simply due to general time-on-task, task difficulty or arousal effects (unlike peak amplitude differences alone, which are notable in the sensory areas).

If motor cortex—including primary motor areas considered to be the main cortical output to motor effectors in the spinal cord[53,54]—are involved in response selection, then where is the neural signal for response execution originating from? To assess whether there were brain regions specifically or preferentially engaged during response execution, we directly contrasted response selection and response execution activity in single-task conditions. The resulting SPMs failed to reveal any brain areas preferentially activated with response execution in either cortical or subcortical tissue (Supplementary Fig. 12). A more parsimonious explanation is that response execution is carried out by the same motor brain regions involved in response selection rather than depending on distinct neural substrates (see "Discussion").

### Granger causality analysis

The picture that emerges from the chronometric results above is that there is a flow of neural information processing from modality-specific perceptual stages to central stages of information processing involved in response selection that consist of both multimodal (MD network) and modality-specific (motor) cortical areas. To test this simple model of flow of information processing, we leveraged the discernable temporal differences in activations across ROIs to determine, with Granger causality analysis, whether there is a causal relationship in activation between these areas.

Granger causality analysis essentially assesses whether activity in a brain region is better predicted by antecedent activity in one or more other brain regions than by its activity alone[55–57]. To carry out this analysis, we constructed a simple bidirectional sensory-MD-motor cortex network using the AO circuit (Fig. 10a), and tested the

hypothesis that Granger causality coefficients would be stronger in that network when participants are performing the AO task than when they are carrying out the VM task (the reciprocal test in the VM network could not be performed because the AO task robustly activates the visual cortex, thus contaminating the "control" condition in the sensory area; see Figs. 2 and 3). Furthermore, based on the hypothesis that perceptual processing precedes processing in the MD network and motor cortex whereas the latter two are co-activated with response selection, we predicted that the auditory cortex ROI would unilaterally "Granger cause" activity in the MD network and motor ROIs, whereas the MD network and oculomotor ROIs would bidirectionally "Granger cause" each other. The results bear out these predictions. As shown in Fig. 10b, the Granger coefficients from the auditory cortex to the MD network, from the auditory cortex to the oculomotor cortex, from the MD network to the oculomotor cortex, and from the oculomotor cortex to the MD network were all stronger in the AO task than in the VM task (from auditory to the MD network, $t(25) = 3.37$, $p = 0.002$, Cohen's $d = 0.66$; from auditory to oculomotor, $t(25) = 4.26$, $p = 0.0003$, Cohen's $d = 0.84$; from the MD network to oculomotor, $t(25) = 2.55$, $p = 0.02$, Cohen's $d = 0.5$; from oculomotor to the MD network, $t(25) = 4.04$, $p = 0.0005$, Cohen's $d = 0.79$). Only the "feedback" connections from the oculomotor cortex to the auditory cortex and from the MD network areas to the auditory cortex failed to show that pattern (from oculomotor to auditory, $t(25) = 0.24$, $p = 0.81$, Cohen's $d = 0.05$; from the MD network to auditory, $t(25) = 0.93$, $p = 0.26$, Cohen's $d = 0.18$). Altogether, the results of the Granger causality analysis in the AO network are consistent with the notion that early perceptual stages of information processing in sensory areas provide the input for the stage of response selection carried out jointly in the MD network and motor cortex areas.

### Discussion

Using ultrafast, high-field fMRI and multivariate analyses, here we tracked the flow of task-specific information processing in simple arbitrary sensorimotor tasks as that information coursed through the brain in the absence (single-task and long-SOA Dual-Task conditions) or in the presence (short-SOA Dual-Task condition) of a competing task. The results were clear cut: Information is processed largely in parallel in the sensory areas, but it hits a central "bottleneck" of information processing once it accesses an ensemble of brain regions known as the multiple-demand (MD) network, where tasks are processed serially. This multimodal network works in tandem with modality-specific motor response areas to define a central bottleneck of information processing at the stage of response selection. The motor response areas are thus not simply response execution areas

that activate "downstream" from the MD network but participate in central information processing.

Classic cognitive models distinguish between parallel streams of information processing at perceptual and motor stages of information processing that are distinct from a central capacity-limited, serial-queuing stage of information processing[5,6,18,45,47]. As a whole, our results provide neural evidence for core aspects of these cognitive models, but also reveal noticeable departures. Processing in the sensory areas temporally preceded (i.e., peaked earlier than) processing in the central MD network and motor areas, as postulated by cognitive models. However, these sensory areas were not entirely functionally spared under dual-task interference. Specifically, when the AO task was presented first but still overlapping with the VM task (Short-SOA Dual-Task condition), activity decoding in the auditory cortex peaked earlier than in single-task or long-SOA dual-task conditions, perhaps to help avoid cross-talk under competing task information[21,22,58–60]. This caveat asides, both our univariate and multivariate results suggest that sensory information processing is functionally distinct from and temporally precede central (MD network) and motor information processing, and that it provides the perceptual input to the central areas[25].

Most importantly, the present findings reveal the neural instantiation of a key aspect of the central bottleneck model[5,6,17,18], namely serial queuing of task-information processing in a specific set of parieto-frontal areas. This ensemble of brain regions corresponds remarkably well to the core MD network, a general-purpose neural network for the encoding and processing of arbitrary task-relevant information[28–30,61]. The empirical identification of the serial bottleneck as the MD network is not without important implications. In particular, since this network is flexibly engaged across multiple cognitive domains to form a domain-general cognitive operating system[61–64], the present results are consistent with the hypothesis that such system's flexibility comes at a cost of limiting the multiplexing of information through that system[21,22].

Interestingly, the MD network may not act alone in forming the central capacity-limited stage of response selection. Rather, both the timecourse and Granger causality analyses suggest that the MD network works in tandem with modality-specific motor areas to select the appropriate input-output associations, alternating which motor area they co-activate with depending on online task demands (see model in Supplementary Fig. 13). These findings are inconsistent with the traditional notion that motor cortical areas, especially primary ones, act primarily downstream from higher-level areas at the response execution stage of information processing[65,66]. Instead, they suggest that motor cortex may be directly involved in the decision-making process[67–70], ostensibly in assisting in the selection of the appropriate motor action. Motor cortex involvement in decision-making is in stark contrast to sensory cortex, whose contribution to the present sensorimotor tasks seems to precede the central stage of response selection. We surmise that it is so because the end product of the computation is a motor response, which should necessarily involve motor areas late in the process, whereas perceptual information could be maintained in the parieto-frontal multiple-demand network[71–73].

Two points about the putative role of motor cortex in the central bottleneck of response selection merit discussion. First, it is likely that the manual and oculomotor areas were involved because our tasks contained a motor response component. Indeed, there is ample evidence that PRP-like dual-task interferences can be obtained even when at least one of the tasks does not include an online motor response[6,11,12,15,16]. Under such circumstances, the central bottleneck may be composed of the conjoint activation of the MD network and the brain areas that subserve these other cognitive process(es), a hypothesis that merits empirical testing. In that respect, this hypothesis shares affinities to "global workspace" neuronal models that attribute conscious perception of a stimulus to the functional interplay between central parieto-frontal areas and task-relevant sensori-motor

areas[74]. In any cases, we do not see the motor cortex as a necessary part of the central bottleneck, but rather as a contributor to it based on the current contextual task demands.

The second point is that a role for motor cortex in the response selection bottleneck does not preclude its involvement in response execution as well. Our finding that the motor areas, or any other areas, were not preferentially activated during response execution is not inconsistent with a dual role of motor cortex in both response selection and execution. After all, motor cortex—especially the primary areas that provide the chief cortico-fugal projections to the spinal cord and elicit fine motor contraction at the weakest stimulation threshold[53,75]—is ideally positioned to elicit motor execution. Moreover, both computational models of decision-making and neurophysiological studies suggest that motor response execution may simply be the end point of a decisional process wherein evidence is accumulated for a specific response(s) until a threshold is reached for execution of that response[48,76–79]. Thus, the most parsimonious account is that motor cortex, both M1 and FEF in the present case, participate in both motor response selection and execution.

The conclusions drawn from the present study were based on results obtained from univariate and multivariate analyses performed on ROIs defined to isolate brain regions distinctly involved in sensory, motor, or central stages of information processing. As such, this approach was successful in testing classic models of information processing and isolating the neural substrates of a serial bottleneck at the central stage of response selection. As mentioned earlier, we identified additional areas to our targeted ROIs with the sensory, motor or conjunction SPMs; namely the superior occipital/posterior parietal cortex, the motor Rolandic cortex, the TPJ and subcortical areas in the thalamus and basal ganglia. However, none of these areas demonstrated clear patterns of activation/decoding in dual-task conditions that would identify them as either modality-specific sensory or motor areas, or central multimodal response selection areas (see Supplementary Figs. 4 and 14 for proposed functions of these areas in the performance of arbitrary sensorimotor tasks). Whatever their functional contributions may ultimately be, they do not seem to take part in the serial amodal bottleneck of information processing that is distinct from early sensory and late response execution stages of information processing.

It's also worth pointing out that while we were able to test cognitive models of the flow of information across sensory, central and motor stages of neural information processing, we could not apply the full set of univariate and multivariate analyses to all the ROIs associated with each of the two tasks. Specifically, cross-task contamination in the visual cortex ROIs and poor decoding in the oculomotor ROI precluded the application of MVPA in dual-task settings in these ROIs. Thus, our results from the latter analyses were primarily drawn from sensory cortex in one task (AO) and motor cortex in the other task (VM). There are two reasons that suggest that this limitation is not a major impediment to the present study's conclusions. First, the two sensory areas and the two motor areas showed similar results in analyses that could be performed on both of them (cf. single-task sensory ROIs and single-task motor ROIs results in Figs. 3, 8 and 9), suggesting that either sensory area and either motor area is a good model of perceptual and motor information processing, respectively. Second, our main conclusions do not rest on this analytical caveat: Serial queuing of information processing in the MD network, motor cortex involvement in response selection (Fig. 9 for both manual- and oculomotor areas), and sensory cortex processing preceding response selection processing (Figs. 8 and 9 for both visual and auditory areas) were all observed using uncontaminated and robust data set.

While our scanning methodology improves the temporal resolution of fMRI by a factor of 10 relative to standard fMRI studies[80], it does not reach the temporal resolution achieved with electrophysiological techniques that can reveal the full extent and complexity of the temporal

dynamics of both feedforward and feedback processing in multitasking conditions[25,26,81–83]. That said, our ultrafast scanning method can temporally discern behaviorally meaningful neural processes in the order of a few hundred milliseconds without sacrificing the high spatial resolution afforded by fMRI. The univariate results in the auditory cortex clearly indicate that a 300 ms shift in activity can be resolved (Fig. 3a), a finding that suggests that cognitive processes unfolding in the order of our temporal sampling (circa 200 ms) should be achievable with the present method. While previous functional neuroimaging studies have explored the application of fast scanning methods to temporally resolve mental processes in the brain, they have generally been obtained under technically limiting conditions (e.g., surface sensors and univariate BOLD responses[7,25,84,85]). The present findings show the potential of using latency-resolved fMRI applied to multivariate data under whole-brain coverage to track task-specific neural activity of cognitive processes as they unfold during task performance. With further improvement in machine hardware, pulse sequencing and analytical methods, it is conceivable that fMRI will one day provide a temporal resolution that might be in the sub-hundred millisecond range[80,86,87]. The combination of such ultrafast imaging techniques, coupled with the relatively high spatial resolution of fMRI, should provide an unprecedented window into our neurobiological understanding of a host of higher-level mental functions, from language comprehension and production to executive control and reasoning.

## Methods

### Participants
Thirty-three adults (19–29 years old, mean: 23 years old; 6 males) participated in the fMRI study. All subjects were recruited from Vanderbilt University community, and received monetary compensation for their participation in the current study. All subjects reported no hearing, neurological or psychiatric disorder. The study procedure was approved by the Vanderbilt University Institutional Review Board, and Informed consent was obtained from all subjects. Four subjects were discarded due to the technical issues with regard to either eye-tracking or button response recording. The data from three additional participants were not included in the analysis because their performances in at least one of the stimulus-response pairings were not different from chance, precluding the isolation of "correct" trials (see below). The results reported here are thus based on data from the remaining 26 subjects (19–29 years old, mean: 22 years old; 6 males).

### Tasks, stimuli, and experimental design
Two distinct sensorimotor tasks were employed in the present study. One task required participants to make an appropriate oculomotor response to an auditory stimulus (AO task), while another task required participants to select an appropriate manual response to a visual stimulus (VM task). For each task, there were eight possible stimuli, each associated with a distinct motor response (i.e., eight Stimulus-Response mappings per task). The auditory and visual stimuli were the same as those used in ref. 7. Briefly, the eight auditory stimuli were highly distinguishable synthetic or natural sounds or complex tones. Each sound was randomly matched to an oculomotor response that consisted in moving the eyes from a central fixation point to one of eight peripheral locations (i.e., up, down, left, right, and four corners; the average eccentricities from the central fixation point in visual angle, were 6.21° for the horizontal locations, 3.74° for the vertical locations, and 7.26° for quadrant locations) that were marked by placeholders on the screen (Fig. 1, placeholders were squares subtending 0.83° of visual angle). The visual stimuli were eight distinct colors presented over the entire display [i.e., yellow (RGB: 255, 255, 30), navy (44, 71, 151), red (237, 32, 36), dark green (10, 130, 65), light blue (79, 188, 220), pink (255, 57, 255), brown (167, 106, 48), and light green (109, 205, 119)]. Each color was

randomly matched to a manual response that consisted in pressing a computer key (during the lab practice session) or a button box key (during the experimental fMRI session) with one of eight possible fingers (thumbs excluded). The auditory and visual stimuli were each presented for 200 msec. At all times, a white fixation dot (with a diameter of 0.2° visual angle) was presented in the center of the screen.

The experiment included both single-task and dual-task trials. In single-task trials, the visual or auditory stimulus was presented for 200 ms, followed by a 2 s window during which the participants' response could be recorded. In dual-task trials, both visual and auditory stimuli were presented with variable stimulus onset asynchrony (SOA) and task order. Specifically, the SOA was either short (300 msec) or long (1500 msec). The short SOA of 300 msec was chosen to ensure that there would be substantial interference between the two tasks while avoiding pitfalls of 0 ms-SOA (e.g., response binding and systematic task prioritization[6,88–90]). The long SOA of 1500 msec was chosen based on previous studies (e.g., ref. 7) showing that this interval was long enough for subjects to complete the first (auditory or visual) 8-alternative-choice task before commencing the second task, thus minimizing dual-task interference. SOA and task order was randomized for each dual-task trial, resulting in four dual-task conditions (i.e., Short-SOA AOVM, Short-SOA VMAO, Long-SOA AOVM, and Long-SOA VMAO). The specific sound-to-oculomotor location (AO task) and color-to-finger (VM task) pairings were randomly assigned for each participant.

Participants were instructed to make the appropriate response to each stimulus as quickly and as accurately as possible independent of the presentation of another stimulus and without delaying either task (if more than one stimulus is presented).

### Pre-scan task learning and practice sessions
Before the fMRI experiment, each participant underwent a single-task learning session and a practice session in the laboratory outside the scanner. MATLAB with Psychtoolbox[91] was used to program the experiment and present the auditory stimuli over headphones and visual stimuli on a computer monitor. Oculomotor responses were recorded using an eye-tracker with ViewPoint software (Arrington Research, Inc.) and manual responses were recorded using a computer keyboard.

The participants first learned the eight stimulus-response pairings for each single task.

For the AO task, each auditory sound was presented with their appropriate oculomotor response indicated by the filled placeholder. After about ten presentations of each S-R mapping during this learning phase (which took about 10 min), participants practiced carrying out the tasks in two blocks of 80 single-task trials, with each S-R pairing repeated 10 times in a random order within a block. For the first of the two practice blocks, trials were self-initiated and performance feedback (in the form of "correct" or "wrong" visual word presentation) was given after each trial. For the second practice block, no feedback was given and each trial was automatically initiated every 5 s.

For the VM task, each of the eight display colors were presented with their appropriate finger response key (as shown on the screen). After about 10 presentations of each pairing during this learning phase (lasting about 10 min), participants performed a practice session which consisted of 2 blocks of 80 trials as described for the AO trials.

By the end of these single-task practice trials, participants had achieved high accuracies for both AO and VM tasks (AO task, mean: 97.4%, SD: 2.2%; VM task, mean: 97.6%, SD: 1.7%). They then performed two blocks of 40 dual-task practice trials. Across these 80 dual-task practice trials, the eight S-R mappings of the AO and of the VM tasks were each presented 10 times (for a total of 80 trials as each dual-task trial includes both tasks), with even presentations of task order and SOA.

After these two dual-task trials-only blocks, participants performed two mixed-trial practice blocks, with each block containing 32 dual-task trials and 16 single-task trials. Specifically, each mixed-trial

practice block included the presentation of each of the 8 S-R pairing for each of the four Dual-Task and two Single-Task conditions (i.e., 8 SR mappings × 6 conditions). Overall, across the single-task and dual-task practice blocks, participants performed 352 single-task trials and 144 dual-task trials. At the end of the practice session, participants with an overall accuracy (i.e., responded to the two tasks correctly and in the correct order) greater than 66.7% in the last two practice (mixed) blocks were invited to participate in the fMRI experiment to secure a sufficient number of correctly-responded trials for fMRI data analysis.

## fMRI procedure and data acquisition

The fMRI experiment was conducted at the Vanderbilt University Institute of Imaging Science. Images were obtained on a Philips 7-Tesla Achieva MRI scanner with a quadrature transmit and 32ch receive head coil. Foam pads were used to keep subjects' heads stabilized during scanning. Functional images were acquired by using a highly accelerated, 3D PRESTO EPI sequence to achieve both short TRs while maintaining BOLD-optimized TEs with short dynamic scan times. The parameters for functional scans were as follow: TR: 199 ms, TE: 27.2 ms, FA: 11°, matrix size: 80 × 80, in-plane voxel size: 2.7 × 2.7 mm². Each scan had 1600 volumes (318.4 s), and for each volume there were 26 axial 4-mm thickness slices to cover the whole brain. An anatomical scan was also acquired for each subject using MPRAGE sequence with the following parameters: TR: 4.8 ms; TE: 2.1 ms, FA: 7°, TI: 1300, shot interval: 4500 ms, 249 sagittal 0.7-mm thickness slices with in-slice matrix size: 352 × 352, voxel size: 0.7 × 0.7 × 0.7 mm³.

The experimental tasks were programmed using Matlab (version: R2018b, MathWorks, Inc.) with Psychtoolbox[91] and back-projected to the participants via a screen board and a mirror attached to the coil. The visual display subtended 16.54° of visual angle. Auditory sounds were presented binaurally to the subjects using an MRI-compatible earphone system (Nordic Neuro Lab) and foam canal tips were used to maximumly reduce scanning noise. An eye-tracking equipment (Avotec, Inc.) mounted on the head coil was used to recorded participant's eyes movement. Eye-tracking calibration and validation were done before scanning. A practice run (with the same trial procedure as the experimental runs) was also administered to each participant while anatomical scans were obtained.

The scanning session consisted of 10 functional runs. Each run had 48 trials, with 8 trials for each of the six conditions (i.e., 4 dual-task conditions and 2 single-task conditions) presented in a random order. Each run included an equal number of the eight S-R mappings for both AO and VM tasks. Thus, across 10 runs, there were a total 320 dual-task trials (2 SOAs × 2 task orders × 8 S-R mappings × 10 runs) and 160 single-task trials (2 tasks × 8 S-R mappings × 10 runs). The inter-trial intervals (ITI) (i.e., from T2 offset to the T1 onset of the next trial) were jittered between 3.98 s and 11.144 s, following a decay distribution with a mean of 5.4 s. This resulted in a duration of 318.4 s (1600 whole-brain volumes) for each run, including a resting-state period of 7.96 s before the first trial to allow equilibrium of the magnetic field, and a resting-state period of 9.96 s at the end of the run to accommodate the hemodynamic delay of the last trial. Most of the participants (22 out of 26) completed 10 functional runs; one participant finished 9 functional runs and three subjects completed 8 functional runs due to technical difficulties.

## Data analysis

**Data preprocessing.** Neuroimaging data preprocessing was performed using AFNI (version: 7.14, Feb. 24, 2021)[92]. First, individuals' anatomical images were skull-stripped using 3dQwarp via @SSwarper. Segmentation was then done on the anatomical image using Freesurfer[93] to obtain masks for white matter, ventricles, and anatomical parcellation of gray matter for each subject. Standard preprocessing (using afni_proc.py)[94] on functional images consisted of: head motion correction, functional-to-anatomical alignment using a local Pearson correlation algorithm (lpc + ZZ cost function in AFNI)[95], whole-brain masking, and time-series local

average scaling for interpreting effect estimates as percent signal change[96]. No spatial smoothing was performed to preserve the spatial variances across neighboring voxels. All analyses were conducted in the subject's native space except for univariate group-level analyses which were performed in a standard space with the functional images being warped and registered to a template (MNI152_T1_2009c) and resampled at 2 × 2 × 2 mm³.

**Univariate analysis.** After preprocessing, several subject-level general linear model analyses and group-level comparisons were performed using AFNI.

First, to isolate sensory and motor regions, a general linear model (GLM1) was applied to each voxel's time series to estimate the auditory and oculomotor BOLD responses to the single AO task, and the visual and manual BOLD responses to the single VM task. The sensory input regressors were modeled by convolving the stimulus onset timings (either auditory or visual) with a double-gamma model of hemodynamic response function. The motor output regressors were modeled by convolving the motor response (i.e., saccade timing and button press timing) with a double-gamma hemodynamic response function. The auditory and oculomotor regressors were modeled on the averaged response to all trials of the single AO task, and the visual and manual motor regressors were modeled on the averaged response to all trials of the single-VM task. The correct and incorrect trials were modeled separately and the subsequent group-level comparisons were performed only on the correct trials. To address the serial correlation issue of fast fMRI data[80,97], we used a generalized least squares time-series fitting approach, with restricted maximum likelihood (REML) estimation of the temporal autocorrelation structure (3dREMLfit with an ARMA model in AFNI). This method has been shown to outperform other approaches in term of providing better task-fMRI reliability[98]. Because multi-shot sequence in ultrafast scanning protocols is motion sensitive and vulnerable to physiological artifacts (cardiac and respiratory), some extra steps were performed to address these issues[97]. Specifically, to reduce the influence of motion-induced artifacts, censoring was applied to the time points in which head motion exceeded a distance of 0.3 mm (i.e., Euclidean norm) over the preceding time point or in which more than 10% of whole brain voxels were regarded as outliers by 3dToutcount[94]. Six head motion correction parameters and six temporal derivatives of head motion were also modeled as nuisance regressors in the general linear model. To address the physiological artifacts, six principal components extracted from white matter areas and six principal components extracted from ventricles were also modeled as nuisance in the general linear model to account for physiological noises[99,100]. The trials of the dual-task conditions were also modeled (as whole-event regressors from stimulus onset of T1) here solely for the purpose of accounting for variance; they weren't used to define the sensory and motor ROIs. All explicit regressors (i.e., single-task and dual-task regressors and any nuisance regressors) were modeled against a fixation baseline that was implicit. These modeling procedures (i.e., independent regressors for correct and incorrect trials, nuisance regressors for motion-correction parameters and physiological noises) were also applied to the other GLMs described further below unless otherwise stated.

The regression coefficients obtained above were used to isolate sensory and motor areas in group-level SPMs of single-task conditions using paired-sample t-tests. Specifically, we contrasted regression coefficients for auditory vs. visual inputs to isolate sensory-specific areas, masked by a contrast of single-task condition relative to the fixation baseline (i.e., auditory input vs. baseline contrasted to visual input vs. baseline). Similarly, we contrasted regression coefficients for oculomotor vs. manual motor responses (each masked by a contrast of single motor response relative to the fixation baseline) to isolate motor-specific areas. Further specificity was added to the manual

motor ROI by including another contrast of left-hand vs. right-hand. We employed a simulation approach to correct for multiple comparisons by using the -Clustsim option of the software tool 3dttest++ (AFNI), as it has been shown to effectively control the false positive rate under 5%[101]. The sensory and motor ROIs were defined based on a threshold at voxel-level of $p < 0.001$, and corrected at cluster-level $\alpha < 0.01$, with cluster size >55 voxels (see Supplementary Table 1).

A separate GLM (GLM2) was used to isolate multimodal central processing regions by searching for brain regions that responded to both single sensorimotor tasks[7,8,12]. This GLM was also used for contrasting dual-task vs. single-task activity (Supplementary Fig. 5). Because the central processing brain regions are expected to be activated for a longer period under dual-task than single-task conditions[7,8,12], we used an analytical approach known to increase the temporal sensitivity of the regression model by first defining subject-specific and condition-specific HRFs[102]. Specifically, based on the signal change time-series from GLM3 (see below), we first extracted the average time-series from an a priori region of interest (i.e., the inferior frontal junction of the Freesurfer anatomical parcellation)[7,8,12] and fitted the average time series with the double-gamma HRF to estimate 5 parameters (i.e., $p1$, $f1$, $p2$, $f2$, and $dip$ as shown in the formula below, with $p1$ and $p2$ representing the time to peak, and $f1$, $f2$ representing the full width at half maximum of the two gamma functions, and $dip$ representing the ratio that adjusts the amplitude of gamma2 relative to gamma1). This procedure was conducted separately for the six task conditions (i.e., 2 ST and 4DT trial-types):

$$\text{HRF}(t) = A \times \left[ \left( \frac{t}{p_1} \right)^{\left( \frac{p_1}{f_1} \right)^2 8 \log(2)} \times e^{-\left( \frac{t - p_1}{f_1^2 / 8 \times \log(2) \times p_1} \right)} - \text{dip} \times \left( \frac{t}{p_2} \right)^{\left( \frac{p_2}{f_2} \right)^2 8 \log(2)} \times e^{-\left( \frac{t - p_2}{f_2^2 / 8 \times \log(2) \times p_2} \right)} \right]$$

(1)

The estimated parameters were then applied in GLM2 to serve as HRF reference for each of the two single-task and four dual-task conditions, thus providing distinct hemodynamic response characteristics for the single-task trials and dual-task trials. As such, GLM2 included the single AO and single VM task trials as the two single-task regressors of interest and the four dual-task trials as the dual-task regressors of interests. Importantly, while the single AO task trials and VM task trials were modeled as single whole events using onset regressors (instead of modeling sensory input and motor output separately as in GLM1), the dual-task trials were modeled as longer-duration, compound events by increasing the $p1$ value in the HRF formula according to its estimation derived from the double-gamma function that curve-fitted the time series extracted from GLM3 in the a priori anatomically-defined IFJ ROI of each subject (see Supplementary Table 3 for $p1$ and other HRF parameter values for all task conditions).

For defining the areas jointly activated by the two tasks (Fig. 2e), their regression coefficients were used in two group-level SPMs: the first consisted of the contrast of single AO trials vs. fixation baseline to find regions activated with the AO task, and the second corresponded to the contrast of single VM task vs. fixation baseline to find regions activated with the VM task. We then conducted a conjunction analysis of these two SPMs to find foci of overlapping activation between AO and VM tasks (i.e., multimodal areas). To compare dual-task to single-task activity (Supplementary Fig. 5), the dual-task regression coefficients were contrasted with those of the single tasks. The conjunction ROIs were defined based on a threshold at voxel level of $p < 0.001$, and corrected at cluster level $\alpha < 0.01$, with cluster size >62 voxels (see Supplementary Table 2), whereas the Dual-Task vs. Single-task SPMs were thresholded at voxel level of $p < 0.001$, and corrected at cluster level $\alpha < 0.01$, with cluster size >44 voxels.

A third GLM (GLM3) was next used to estimate the time series of signal change in the sensory, motor and central ROIs for the six

experimental conditions. Specifically, a multiple-parameters shape-free HRF model (i.e., TENT function in AFNI) was used for each of the six task condition regressors of interest to estimate the percent signal change at each time point from stimulus onset (T1 stimulus onset in the dual-task trials) to 14 s later with a TR time step (i.e., 199 ms). Thus, there were 69 estimations characterizing the temporal profile of signal change time series spanning the whole trial for each of the single- and dual-task conditions. The signal change time series were extracted from an ROI (e.g., sensory, motor, multimodal ROIs obtained from GLM1, GLM2) and were averaged across all voxels in that ROI for each experimental condition. This average signal change time series were then curve-fitted using a data-driven smoothing spline approach[103]. Based on the fitted curves, the onset latency, peak latency, and peak amplitude were estimated[84,104,105]. The peak latency was defined as the time at which the curve had reached its peak amplitude. The onset latency was defined as the time at which the curve reached 10% of its maximal amplitude. The parameter estimations were performed at the individual level, and the estimated parameters were then submitted to group-level comparisons. Specifically, to test a task effect for a given parameter of the BOLD profile (e.g., peak latency), an omnibus ANOVA was conducted on the means of that parameter across task conditions of interest, followed by paired sample $t$-tests when applicable.

Two additional GLMs were carried out for the purpose of multivariate analysis (see next section). The first, GLM4, was constructed to estimate activation for each individual trial (i.e., each S-R mapping) of each of the two single-task conditions. This was done by including, for each trial, a whole-event regressor modeled from trial onset and convolved with a classic double-gamma HRF model. Because each single-task trial has its own S-R mappings, GLM4 yielded beta values (regression coefficients) for each of the 8-alternative S-R mapping trials for each of the AO and VM tasks. These beta values were used to train task-specific classifiers for both single-task and dual-task multivariate decoding analyses (see below).

Finally, GLM5 was performed to estimate the signal change time series of the multivariate analyses for each trial of each of the six task conditions. To ensure sufficient power for this trial-level modeling, a Least-Squares-Sum approach was used (3dLSS function in AFNI)[106]. Specifically, for a given task condition, a set of 69 regressors (one per time point) spanning from stimulus onset to 14 s later (i.e., a time-resolved HRF-free procedure similar to GLM3) was modeled for a single trial, and a second set of 69 regressors was modeled for the averaged response to all other trials for that condition. Thus, there were only two sets of regressors modeled for a given task condition in this regression model. This procedure was repeated for each single trial separately, which gave an estimate (of the signal change time-course) for each trial for that task condition. The same GLM5 approach was conducted separately for the other five task conditions, thus yielding signal change timecourses for all trials for each of the six task conditions. Only estimations for correct trials were used for single-task and dual-task multivariate decoding analysis.

**Single-task multivariate decoding analysis.** MVPAs were performed in MATLAB with the Princeton MVPA toolbox. Classification analyses were conducted in ROIs defined from the univariate analyses; that is, modality-specific sensory, motor and multimodal central processing ROIs.

For the single-task MVPA analyses, we sought to determine if the sensory, motor, or multimodal ROIs would show task-specific activation patterns. For the sensory and motor ROIs, features selection was conducted by choosing the 50 most activated voxels (i.e., top $t$-values from individual subject GLM) in their respective condition relative to the baseline[43]. For the multimodal ROIs, feature selection was conducted by selecting the 50 most co-activated voxels in a conjunction between the two single tasks. A classifier was trained to discriminate among 8 alternative choices for each single task (e.g., 8 auditory-oculomotor

mappings for the AO task). Specifically, we used the estimations (beta values) from GLM4 to train a task-specific classifier to discriminate one S-R mapping against the other 7 S-R mappings using L2-regularized logistic regression algorithm and set the penalty value at 25[107,108]. For testing, the trained task-specific classifier was applied to the corresponding task trials by using a leave-one-out cross validation procedure to ensure the independence of the training and testing dataset. A trial was considered correctly classified if the decoded S-R mapping matched the actual one. Classification accuracy was scored by counting the proportion of correctly classified trials. The trained task-specific classifier was also applied to the other (i.e., control) task trials using a train-data1-test-data2 procedure to validate the specificity of the trained classifier. Significance was assessed at the group-level by comparing classification accuracy against a chance level (12.5%) by using a one-tailed (right) $t$-test, and corrected for multiple comparisons[44] in each ROI.

For any given sensory or motor ROI, we expect to see successful decoding of trials that correspond to the task that classifier was trained on (e.g., successful AO decoding of AO trials in auditory cortex ROI), but not for trials of the other task (e.g., no AO decoding of VM trials in auditory cortex ROI) nor for the decoder of a different sensory/motor modality (e.g., no VM decoding of AO or VM trials in auditory cortex ROI). Since the multimodal central processing ROIs are activated by both single tasks, a different decoding pattern is expected. Specifically, we should observe successful decoding for both of the task-specific classifiers, but only for their corresponding trials (e.g., successful AO decoding of AO trials but not VM trials, and successful VM decoding of VM trials but not AO trials).

**Time-resolved multivariate decoding analysis.** In order to temporally resolve task-specific activity in not only the single-tasks but especially in the dual-task conditions, the trained task-specific classifiers were also applied to trace the timecourse of decoding in each task condition. That is, the trained AO- or VM-task classifiers were used to decode the S-R mappings at each time point of the signal change timecourse estimated with GLM5 for the single-task and dual-task trials. For the single-task trials, the leave-one-out cross validation procedure was adopted to ensure the independence of the testing data (i.e., the left-out timecourses of single-task trials from GLM5) from the training data (i.e., beta values of all remaining single-task trials from GLM4). For the dual-task trials, the training data (single-task trials) and testing data (dual-task trials) were independent (i.e., train-data1-test-data2 procedure) so there was no need for the leave-one-out procedure. Classification accuracy was scored in the same way as in the single-task decoding analyses but here at each time point of the timecourses of both the single-task and dual-task trials, resulting in task-specific classification accuracy timecourses for each task conditions. The classification accuracy timecourses were then curve fitted with a single-gamma HRF rather than a smoothing spline function due to the noisier data[109], and the classification peak latency and onset latency were estimated from the fitted curve. The decoding time courses for a given task under single-task and dual-task conditions were then overlaid for comparisons, making sure to take into consideration for the dual tasks the proper task order and to time-shift the Task2 decoding timecourse by the duration of the long or short SOA so that all task conditions are appropriately time-aligned. To improve decoding timecourse SNR in the MD network ROIs, we combined task order across conditions (e.g., Task1 decoding timecourse in Fig. 7a is the average of AO and VM decoding timecourses from the respective AOVM and VMAO dual-task conditions) after verifying in an ANOVA with factors of target (T1/T2), task order (VM-AO/AO-VM) and SOA(300 ms/1500 ms) that there were no effect of task order ($ps >$ 0.10). The decoding peak latency and onset latency were submitted to group-level analyses. Significance was assessed by comparing the decoding onset/peak latencies between dual-task and single-task conditions (e.g., dual-task T2 vs. single-task) and between SOAs of

dual-task conditions. In order to assess the behavioral relevance of the observed onset and/or latency differences, we carried out an individual difference analysis that correlated the decoded latency data to the magnitude of participants' PRP (i.e., RT2 difference between short and long SOA in dual-task conditions).

**Single-task quartile RT analysis.** For each participant, trials of each of the two single-task conditions were binned into four quartiles according to their RTs. A general linear model (GLM6) was constructed to estimate the signal change time series for each single-task quartile condition. The modeling procedures were the same as GLM3 except that four quartile regressors instead of one were modeled for each single-task condition. The GLM6s were conducted for the AO and VM tasks separately. The signal change time series curve fitting and parameter estimation were the same as in the univariate BOLD time series analysis. Group-level analysis (e.g., ANOVA) was conducted to compare difference(s) among quartile conditions in the sensory, motor and multimodal ROIs.

**Granger causality analysis in single tasks.** Granger causality (GC) analyses were conducted using the Matlab toolbox MVGC[110]. The single-task time series for all trials (GLM5) in the AO neural circuit (auditory cortex, MD network, and FEF ROIs) were used in the GC analyses. The order of the vector autoregressive (VAR) model was estimated using the Akaike information criterion (AIC). Then the VAR model was estimated for the selected model order, and pairwise-conditional Granger causalities was calculated in time-domain from VAR model parameters by state-space method[111]. For each pairwise connection, Granger causalities were compared between AO task and VM task conditions.

**Reporting summary**
Further information on research design is available in the Nature Portfolio Reporting Summary linked to this article.

## Data availability
The raw neuroimaging data supporting the conclusions of this study are available at https://s3.accre.vu:9000/maroislabbucket/maroisnatcomms/prp_raw_data.zip. Source data are provided with this paper.

## Code availability
Scripts for neuroimaging preprocessing and univariate analyses and custom MATLAB scripts for MVPA are available at https://s3.accre.vu:9000/maroislabbucket/maroisnatcomms/prp_scripts.zip.

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

## Acknowledgements

The research reported here was supported by a fund from National Eye Institute of National Institutes of Health (NIH (NEI): 5P30EY008126-33), and a scientific research start-up fund for high-end talents from Shenzhen University awarded to Q.Y. We thank Vanderbilt University Institute of Imaging Science and Michael Budianto for their help with fMRI data collection. We also thank Gang Chen for his helpful discussions on data analyses.

## Author contributions

R.M. and Q.Y. conceived the study and designed the experiment; A.T.N. developed the fMRI scanning protocol; Q.Y. carried out the experiment, collected the data, and performed the data analyses; Q.Y. and R.M. interpreted the results, R.M. and Q.Y. wrote and revised the manuscript; R.M. supervised the study.

## Competing interests

The authors declare no competing interests.
