## [Transparent Peer Review file · Nature Communications]

Ultrafast fMRI reveals serial queuing of information processing during multitasking in the human brain

Corresponding Author: Dr Qiu Hai Yue

Version 0:

Reviewer comments:

Reviewer #1

(Remarks to the Author)

This is a very good paper. Using 7T imaging at TR=199 ms, and basing their question on a sophisticated and detailed cognitive model, the authors have extracted some telling findings concerning stimulus-response mapping and the central processing bottleneck in the brain. Most importantly, multivariate analysis shows that two closely-successive tasks are queued in the multiple-demand network, while this queuing is preceded by independent, parallel operations in the sensory systems called for by the two tasks. These results provide a compelling brain-based explanation for a classic finding from the psychological literature. There are also a good number of further nice results. The two single tasks produce separate and shared regions of brain activity, with the expected patterns of decoding (Figure 5) and decoding latencies (Figure 9). Delay of one task by just 300 ms produces the expected latency difference in its initial sensory response (Figure 7), showing that the ultrafast imaging does its job. A useful supplementary finding is the involvement of motor cortex at the response selection stage of the task, indicating that there is no simple feedforward process from multiple-demand to motor regions.

The analysis is sophisticated and comprehensive, and the presentation excellent. I have just minor suggestions for clarification:

p. 18. Instructions concerning speed and required order of response (if any) could be stated. I struggled with line 734, which could perhaps be extended to explain that the 10 repetitions of each S-R pairing were not crossed with condition (and combined over blocks).

Line 834. Was the duration of each response modelled, or if not, was this just an onset regressor? Baseline was implicit after removing all explicit regressors, I presume?

Lines 882-891. For this time-resolved decoding, I think there was just one classifier used for all time points (unlike dynamic coding models), but this could be stated.

Several times, "insula" is written as "insular".

Figure 4. Aren't the two MD panels identical?

Figures 7 and 8. In left columns, why are shifts 200 and 1400 instead of 300 and 1500?

Reviewer #2

(Remarks to the Author)

In this paper, the authors investigated processing bottlenecks in the human brain. From behavioral studies and cognitive modeling, it is well known that such bottlenecks exist; the current paper uses an elegant experimental design in combination with ultrafast fMRI to track where in the brain they occur. The results confirm consisting psychological theories of

multitasking. The most surprising result, although it has also been observed in other studies, is the involvement of the motor system in response selection, in contrast to response execution. In addition, the manuscript is clearly written, and the analyses were cleverly designed and follow a logical setup throughout. I have no remarks, and suggest the paper to be accepted in its current form.

Reviewer #3

(Remarks to the Author)

Yue and colleagues present an interesting fMRI study into the neural sequelae of sensory, central, and motor processing under single vs. dual task conditions (at long vs. short ISI). Using highly time-resolved fMRI and neural classification analyses allows the authors to trace task-specific processes first in independent sensory regions, and then in their serial “cuing” during a processing bottleneck associated with multiple-demand and motor regions. The results essentially confirm long-held assumptions about the transition from stimulus processing to response selection and execution, though a surprise is that motor regions seem to also be implicated in response selection rather than execution alone.

This study is well-designed and technically sophisticated, and the paper is written in a reasonably accessible manner. The question of the source of dual-tasking limitations in human cognition is also an important one. While it could be argued that many of the findings reported simply confirm known or assumed facts about the brain (e.g., there is no plausible null hypothesis about the sequence of activation – it will obviously begin in sensory cortex and end in motor cortex!), in my view, the combination of the sophisticated approach and some novel findings is just about sufficient to justify publication in an outlet like Nature Communications. I only have a few comments that will hopefully aid in making this paper even better. My comments are not ordered by importance.

1- Given that the Methods section is presented at the end, I think it would be good to inform the reader a little bit more about the nature of the contrasts/modeling of events in the results section, too. Specifically, it would be good to note explicitly the event onsets and durations of the model regressors for the different conditions, so that it is more transparent that e.g. the dual task conditions were modeled as longer-duration, compound events, whereas the single-task conditions are not. Speaking of the latter, could you also be more explicit in the Methods section on the duration of these compound events in the GLM models?

2- If I understand it correctly, the key results in the dual-task condition analyses end up involving sensory cortex data derived from the AO task only and motor cortex data from the VM task only. Could this impose some limitations on the interpretation/generalization of the results? This might merit some caveats or discussion.

3- My main conceptual point concerns the somewhat equivocal findings for MD and motor regions in their putative involvement in “central” response selection processes. I wonder whether one reason for this surprising result is that the type of tasks employed here essentially involve well-learned S-R associations, or “flat-mapping” of responses onto stimuli. This set-up is understandable in that these simple and well-learned SR tasks are typical for PRP studies, but it should also be noted that these tasks lend themselves to becoming quite automatized, meaning that presentation of a given stimulus likely leads to a direct mnemonic retrieval of the associated response, and does not require much in the way of “algorithmic” central cognitive processing in between sensory analysis and motor response. This sort of task might therefore promote more overlap (both temporal and functional) between MD regions and motor regions than would tasks that require more cognitive deliberation for response selection. In my view, this deserves some discussion that could contextualize the findings of motor cortex seemingly doing equivalent things as the MD regions in this protocol. Come to think of it, it might be interesting to analyze whether subjects incurred switch costs when moving from AO to VM trials (and vice versa), and those trials, where at least a change in SR mapping is required, might be associated with a more distinct neural profile between MD and motor regions as well. (Just an idea).

Version 1:

Reviewer comments:

Reviewer #1

(Remarks to the Author)

The authors have responded carefully to my (minor) questions, and I am happy to recommend publication.

Reviewer #3

(Remarks to the Author)

In my opinion, the authors did a thorough job of addressing all of the reviewers' concerns. This is a strong paper ready for publication.

General Note to Reviewers: To conform to *Nature Communications'* formatting guidelines that limit the number of figures to 10, Fig. 1 and Fig. 12 of the original manuscript now appear in the Supplementary Information (as Supplementary Figures 1 and 14, respectively).

Reviewer #1 (Remarks to the Author):

This is a very good paper. Using 7T imaging at TR=199 ms, and basing their question on a sophisticated and detailed cognitive model, the authors have extracted some telling findings concerning stimulus-response mapping and the central processing bottleneck in the brain. Most importantly, multivariate analysis shows that two closely-successive tasks are queued in the multiple-demand network, while this queuing is preceded by independent, parallel operations in the sensory systems called for by the two tasks. These results provide a compelling brain-based explanation for a classic finding from the psychological literature. There are also a good number of further nice results. The two single tasks produce separate and shared regions of brain activity, with the expected patterns of decoding (Figure 5) and decoding latencies (Figure 9). Delay of one task by just 300 ms produces the expected latency difference in its initial sensory response (Figure 7), showing that the ultrafast imaging does its job. A useful supplementary finding is the involvement of motor cortex at the response selection stage of the task, indicating that there is no simple feedforward process from multiple-demand to motor regions.

The analysis is sophisticated and comprehensive, and the presentation excellent. I have just minor suggestions for clarification:

We thank the reviewer for their positive comments about our study. Below, we address each of their comments.

p. 18. Instructions concerning speed and required order of response (if any) could be stated. I struggled with line 734, which could perhaps be extended to explain that the 10 repetitions of each S-R pairing were not crossed with condition (and combined over blocks).

We apologize for the omission of task speed/order instructions. We now include this information on p. 19 of the revised manuscript. We also agree the sentence on line 734 could have been clearer. We have now clarified this sentence to state that across the 80 dual-task practice trials, the eight S-R mappings of the AO and VM tasks were presented 10 times (for a total of 80 trials as each dual-task trial includes both tasks), with balanced presentations of the two task orders and the two SOAs across these 80 trials (see p. 19).

Line 834. Was the duration of each response modelled, or if not, was this just an onset regressor? Baseline was implicit after removing all explicit regressors, I presume?

Regarding GLM2 as described in Line 834, only onset regressors were used to model the single-task trial responses. We now mention this on p. 23 of the revised manuscript. The main purpose of GLM2 was to isolate the candidate regions for central processing based on overlap of the two single-task activations. The variability in response times was not considered at that stage (though it was of course considered at later analytical stages; e.g. see GLM3). Note that our approach was successful in isolating several areas of overlap, only a subset of which showed the characteristics of a central bottleneck, as we mention in the paper (on pp. 15-16). We should add that GLM2 was also used to contrast dual-task to single-task activity (Fig. S5 in Supplementary Information), and for this model the dual-task regressors were of longer duration than the single-task regressors, as one would expect (see pp. 22-23).

Regarding the baseline, the reviewer is correct; all explicit regressors (i.e., single-task and dual-task regressors and any nuisance regressors such as head motion and physiological noises) were modelled against a fixation baseline that was implicit. We now make that clear on p. 22 of the revised manuscript.

Lines 882-891. For this time-resolved decoding, I think there was just one classifier used for all time points (unlike dynamic coding models), but this could be stated.

The reviewer is correct; there was just one classifier used for all time points of the time-resolved decoding analysis. Specifically, we used the beta values estimated across all time points for a single-task trial (from GLM4) to obtain a reliable estimate of single-task activation and then used these estimates to train task-specific classifiers. These trained classifiers were then used to decode the time points of the activation timecourses of dual-task trials as explained in the 'Single-task multivariate decoding analysis' section of the Methods (p. 24). We now explicitly state that we used single-task classifiers on pp. 23-24 of the revised text.

Several times, "insula" is written as "insular".

This pesky auto-completion error (and our oversight of this error) has now been corrected.

Figure 4. Aren't the two MD panels identical?

They are. We duplicated the MD panel so that the reader could more easily compare the activation timecourses in the MD network relative to the sensory and motor areas for each (AO/VM) task. We now make that point clearer in the legend of that figure (now corresponding to Fig. 3). We thank the reviewer for pointing this out.

Figures 7 and 8. In left columns, why are shifts 200 and 1400 instead of 300 and 1500?

We are grateful to the reviewer for picking this up. The raw timecourses in the left panels could only be shifted by multiples of TRs to the nearest SOAs (199ms and 1393ms, which we rounded up to 200ms and 1400ms) whereas the curve-fitted data could be shifted by the exact SOAs (300ms and 1500ms). Of course, the statistical inferences were calculated on the curve-fitted data. We have now added the exact numbers in the figure and clarified the time-shift differences between the raw and curve-fitted data in the legends of what is now Fig. 6 and 7.

Reviewer #2 (Remarks to the Author):

In this paper, the authors investigated processing bottlenecks in the human brain. From behavioral studies and cognitive modeling, it is well known that such bottlenecks exist; the current paper uses an elegant experimental design combined with ultrafast fMRI to track where in the brain they occur. The results confirm existing psychological theories of multitasking. The most surprising result, although it has been observed in other studies, is the involvement of the motor system in response selection in contrast to response execution. In addition, the manuscript is clearly written and the analyses were cleverly designed and follow a logical setup throughout. I have no remarks and suggest the paper to be accepted in its current form.

We thank the reviewer for their positive comments about our study.

Reviewer #3 (Remarks to the Author):

Yue and colleagues present an interesting fMRI study into the neural sequelae of sensory, central, and motor processing under single vs. dual task conditions (at long vs. short ISI). Using highly time-resolved fMRI and neural classification analyses allows the authors to trace task-specific processes first in independent sensory regions, and then in their serial “cuing” during a processing bottleneck associated with multiple-demand and motor regions. The results essentially confirm long-held assumptions about the transition from stimulus processing to response selection and execution, though a surprise is that motor regions seem to also be implicated in response selection rather than execution alone.

This study is well-designed and technically sophisticated, and the paper is written in a reasonably accessible manner. The question of the source of dual-tasking limitations in human cognition is also an important one. While it could be argued that many of the findings reported simply confirm known or assumed facts about the brain (e.g., there is no plausible null hypothesis about the sequence of activation – it will obviously begin in sensory cortex and end in motor cortex!), in my view, the combination of the sophisticated approach and some novel findings is just about sufficient to justify publication in an outlet like Nature Communications. I only have a few comments that will hopefully aid in making this paper even better. My comments are not ordered by importance.

We thank the reviewer for their positive comments about our study. Below, we address each of their comments.

1 - Given that the Methods section is presented at the end, I think it would be good to inform the reader a little bit more about the nature of the contrasts/modeling of events in the results section, too. Specifically, it would be good to note explicitly the event onsets and durations of the model regressors for the different conditions, so that it is more transparent that e.g. the dual task conditions were modeled as longer-duration, compound events, whereas the single-task conditions are not. Speaking of the latter, could you also be more explicit in the Methods section on the duration of these compound events in the GLM models?

We realize our Methods could have been more clearly stated, and we have worked on clarifying it in the revised manuscript. Following the reviewer’s helpful suggestion, we have expanded the text at select parts of the Results to briefly introduce each of the five GLMs and highlight the distinctive characteristics of their modeling events and contrasts. We have also added text on pp. 22-23 of the Methods section about the duration of the compound events in some of these GLM models. Essentially, we mention in that section that while the single AO task trials and VM task trials were modeled as single whole events, the dual-task trials were modeled as longer-duration, compound events in GLM2 by increasing the p1 value (i.e. time-to-peak) in the HRF formula. The p1 values were derived from the double-gamma functions that curve-fitted the time series extracted from the *a priori* anatomically-defined IFJ ROI of each subject in GLM3. We have also added a Table in Supplementary Information (Supplementary Table 3) that lists the group-averaged parameters of the HRF formula derived in this manner for all task conditions.

2 - If I understand it correctly, the key results in the dual-task condition analyses end up involving sensory cortex data derived from the AO task only and motor cortex data from the VM task only. Could this impose some limitations on the interpretation/generalization of the results? This might merit some caveats or discussion.

The reviewer correctly points out that comparison of the decoding timecourses in the MD network to the sensory (auditory cortex) and motor (manual cortex) areas was not fully crossed (meaning it did not include visual cortex and oculomotor cortex because of either cross-task contamination [in the visual cortex] or poor decoding [in

oculomotor cortex]). We note, however, that the two sensory areas showed similar results in the analyses that could be performed on both of them – that is, whenever the visual sensory ROI was not contaminated by the oculomotor task (cf. single-task results in auditory and visual ROIs in Figs. 3, 8 and 9) – which suggests that either sensory area is a good model of perceptual processing. The same logic applies to the two motor areas: i.e. they behaved similarly to one another when SNR was not a limiting issue (cf. single-task results in Figs. 3, 8, and 9). It is also important to point out that the main conclusions of the paper do not depend on having a fully-crossed analysis: Serial queuing of information processing in the MD network; evidence that the motor cortex is involved in the bottleneck of response selection (Fig 9 for both manual- and oculo-motor areas); and sensory cortex activity preceding and being largely unaffected by the central bottleneck of information processing (Figs 3, 8 and 9), were all conclusions drawn from uncontaminated and robust data sets. Nevertheless, we now mention this noteworthy caveat on p. 16 of the discussion of the revised manuscript.

3 - My main conceptual point concerns the somewhat equivocal findings for MD and motor regions in their putative involvement in “central” response selection processes. I wonder whether one reason for this surprising result is that the type of tasks employed here essentially involve well-learned S-R associations, or “flat-mapping” of responses onto stimuli. This set-up is understandable in that these simple and well-learned SR tasks are typical for PRP studies, but it should also be noted that these tasks lend themselves to becoming quite automatized, meaning that presentation of a given stimulus likely leads to a direct mnemonic retrieval of the associated response, and does not require much in the way of “algorithmic” central cognitive processing in between sensory analysis and motor response. This sort of task might therefore promote more overlap (both temporal and functional) between MD regions and motor regions than would tasks that require more cognitive deliberation for response selection. In my view, this deserves some discussion that could contextualize the findings of motor cortex seemingly doing equivalent things as the MD regions in this protocol. Come to think of it, it might be interesting to analyze whether subjects incurred switch costs when moving from AO to VM trials (and vice versa), and those trials, where at least a change in SR mapping is required, might be associated with a more distinct neural profile between MD and motor regions as well. (Just an idea).

The reviewer makes an incisive point, which we address in a three-pronged manner.

First, we wish to specify that while S-R tasks of the type we used here are common in PRP experiments, our tasks were not automatized. The high number of alternative response mappings (two 8 alternative-discrimination tasks) and relatively little practice the participants were exposed to ensured that these tasks required attention and cognitive effort. The long single-task RTs (about 1 second) are a clear manifestation of that effort. Moreover, the pronounced dual-task interference is another indication that the tasks were not carried out automatically, as parallel task processing is one of the hallmarks of automatized behavior (e.g. Logan, *Psyc Rev.*, 1988). (Note, however, that we are not arguing that RS tasks cannot be automatized; there is evidence that it can happen with simple 2 AD tasks after thousand trials of practice; e.g. Schumacher et al., *Psyc Sci*, 2001).

That said, we believe the reviewer is absolutely correct in pointing out that the proposed involvement of the motor areas in the central bottleneck (along with the MD network) could very well be related to the fact that the tasks involved a motor response component. Cognitively demanding tasks need not be associated with a motor response, and indeed there is ample evidence that a PRP can be obtained even when at least one of the tasks does not have an online motor response (e.g., Tombu et al., *PNAS*, 2011). We realize that our original manuscript didn't make that point explicit, and we have now done so in the revised manuscript (see p. 15), including the abstract.

Finally, the reviewer suggests looking at task-switching situations in our data to see if the role of the MD and motor areas can be teased apart in the central bottleneck. While we agree that looking at task switching costs would be interesting, we believe it is beyond the scope of the present study, especially since we are already confronted with the space limitations afforded in *Nature Communications*. More importantly, from a conceptual standpoint, looking at task switching is not an ideal manner to address the reviewer's point because the task

switch here is not devoid of a motor component as it consists in switching between two task rules that contain motor responses. Moreover, isolating task-switch costs is not straightforward in our experimental design because we do not have the ideal control condition, which would be dual-task trials in which both Task 1 and Task 2 were of the same modality pair (i.e. a dual-task AO-AO trial or VM-VM trial) to compare with the task-switch conditions (AO-VM or VM-AO). We believe that the best way to address the reviewer's point is to have an experimental design in which at least one or both of the tasks do not involve online motor responses. We are currently carrying out such experiment using an attentional blink-type design in which two temporally overlapping tasks reveal a deficit in the conscious perception of the second of two targets (as opposed to a delay in response time of the second of two targets as in the PRP). Our working hypothesis in this case is that the central bottleneck will consist of the conjoint activation of the MD network with perceptual areas but not with motor areas. We shall see.